# Stretchable OLEDs based on a hidden active area for high fill factor and resolution compensation

Donggyun Lee [1], Su-Bon Kim [1], Taehyun Kim[1], Dongho Choi [1], Jee Hoon Sim [1], Woochan Lee [1], Hyunsu Cho [2], Jong-Heon Yang [2], Junho Kim [1], Sangin Hahn [1], Hanul Moon [3] ✉ & Seunghyup Yoo [1] ✉

Stretchable organic light-emitting diodes (OLEDs) have emerged as promising optoelectronic devices with exceptional degree of freedom in form factors. However, stretching OLEDs often results in a reduction in the geometrical fill factor (FF), that is the ratio of an active area to the total area, thereby limiting their potential for a broad range of applications. To overcome these challenges, we propose a three-dimensional (3D) architecture adopting a hidden active area that serves a dual role as both an emitting area and an interconnector. For this purpose, an ultrathin OLED is first attached to a 3D rigid island array structure through quadaxial stretching for precise, deformation-free alignment. A portion of the ultrathin OLED is concealed by letting it 'fold in' between the adjacent islands in the initial, non-stretched condition and gradually surfaces to the top upon stretching. This design enables the proposed stretchable OLEDs to exhibit a relatively high FF not only in the initial state but also after substantial deformation corresponding to a 30% biaxial system strain. Moreover, passive-matrix OLED displays that utilize this architecture are shown to be configurable for compensation of post-stretch resolution loss, demonstrating the efficacy of the proposed approach in realizing the full potential of stretchable OLEDs.

Stretchable organic light-emitting diodes (OLEDs), capable of bending, twisting, and conforming to curved surfaces[1], have emerged as innovative optoelectronic devices for next-generation displays and lighting applications. Their exceptional form factor advantages have propelled them beyond the traditional use of rigid or flexible OLEDs, poised to open up diverse possibilities for non-conventional applications such as wearable displays as well as body-attachable health-monitoring[2–6] or phototherapeutic devices[7–9]. The automotive industry may also benefit from this technology, as it offers opportunities to create a new generation of in-car displays with enhanced safety and decorative features[10–12]. With its cutting-edge form factor

and versatile applications, stretchable OLEDs have captured the attention of the organic electronics field in pursuit of the ultimate form factor[13–15].

Recent studies have grouped the ways to make stretchable OLEDs into three main types: first, by creating intrinsic stretchable materials that can be used directly in devices[16–24]; second, by transferring thin OLEDs onto stretched substrates and then releasing the strain to form an accordion-like wavy surface[25–31]; and lastly by using arrays of rigid islands connected with stress-relieving interconnectors[32,33]. Among these approaches, the last one is considered to be the most promising. This is because it allows for the use of conventional high-performance

[1]School of Electrical Engineering, Korea Advanced Institute of Science and Technology (KAIST), Daejeon 34141, Republic of Korea. [2]Electronics Telecommunications Research Institute (ETRI), Daejeon 34129, Republic of Korea. [3]Department of Semiconductor; Department of Chemical Engineering (BK21 FOUR Graduate Program), Dong-A University, Busan 49315, Republic of Korea. ✉e-mail: hmoon@dau.ac.kr; syoo@ee.kaist.ac.kr

organic materials, electrodes, and multi-layer encapsulation technology. In this method, the active area is confined to rigid sections, with interconnectors bearing the stress. This setup allows the active area to be sealed with materials such as alumina ($Al_2O_3$) and nitrides, which are one of the best in gas barrier properties yet have relatively low crack-onset strain (COS). However, there is a main challenge to overcome. For display applications, the presence of stress-relieving interconnectors (e.g., serpentine-shaped interconnectors) limits the overall resolution or pixel density, and stretching can further reduce resolution, potentially deteriorating image quality. Even for non-display uses, the limited initial and post-stretch fill factor (FF), i.e., the proportion of the active area to the entire surface area before and after stretching, could be an issue. For instance, if used for a wearable phototherapeutic patch, the limited FF could result in some areas of the skin being left unilluminated.

In this work, we propose a solution to achieve rigid-island-based stretchable light sources with the initial and post-stretch FF that are both unprecedentedly high. To this end, we exploit the high degree of mechanical flexibility and areal light-emitting characteristics of ultra-thin OLEDs in a three-dimensional structure. Specifically, we replace serpentine interconnectors with an ultra-thin hidden active area (HAA). It is hidden in the initial unstretched state by 'folding inward' along the negative $z$-axis between adjacent islands of a 3D rigid island array at a very tight bending radius, thereby allowing a very high FF to be achieved in the initial, unstretched state. This ultrathin HAA emerges to the surfaces upon stretching, alleviating the typical sharp drop in FF upon stretching or, optionally, offering a means to compensate stretching-induced resolution decrease. Unlike previous approaches adopting compact initial integration structures[34,35], hidden interconnectors[36–38], or hidden active devices[39–41], the HAA in the present work is a part of a whole ultrathin OLED transferred to the islands, and thus, its full area can ultimately function as an active light-emitting

region, enabling very high FF even in the stretched state. We describe a fabrication process to integrate ultrathin OLEDs onto the 3D array structure, along with a scheme for precise alignment between these two elements. In particular, we adopt a quadaxial stretching method[42] that ensures distortion-free alignment, addressing the shortcomings of conventional biaxial stretching techniques. To implement the HAA concept in a fail-safe manner, we have carefully designed the device structure and dimensions, using mechanical simulations based on finite element method (FEM). This ensures minimal strain on the 3D island arrays as well as that on the encapsulation layers and electrode layers of the ultrathin OLED. The resulting stretchable OLEDs not only exhibit a high initial FF (97%), but also maintain their FF up to 87% after experiencing significant biaxial deformation with a system strain of 30%. This is a level of performance unattainable in conventional 2D rigid-island configurations. Furthermore, these high FF stretchable OLEDs demonstrate robustness, with a decrease of only about 10% in current efficiency even after 1000 biaxial stretching cycles under ca. 30% biaxial system strain. Based on this mechanical reliability, we demonstrate the versatility of these stretchable OLEDs by showcasing their function on various curved surfaces, including spheres, cylinders, balloons, and human body surfaces. Finally, we demonstrate a passive matrix (PM) array utilizing HAA as hidden pixels, capable of compensating for post-stretch resolution reduction inherent to conventional stretchable displays.

## Results

### Overview of the proposed stretchable OLEDs

Figure 1a provides a schematic diagram of the conventional rigid island arrays with stress-relieving interconnectors platform (conventional platform) in its initial (i) and stretched state (ii), and the proposed hidden active area (HAA) structure in its initial (iii) and stretched state (iv). To calculate the system strain, we define the length of the island as

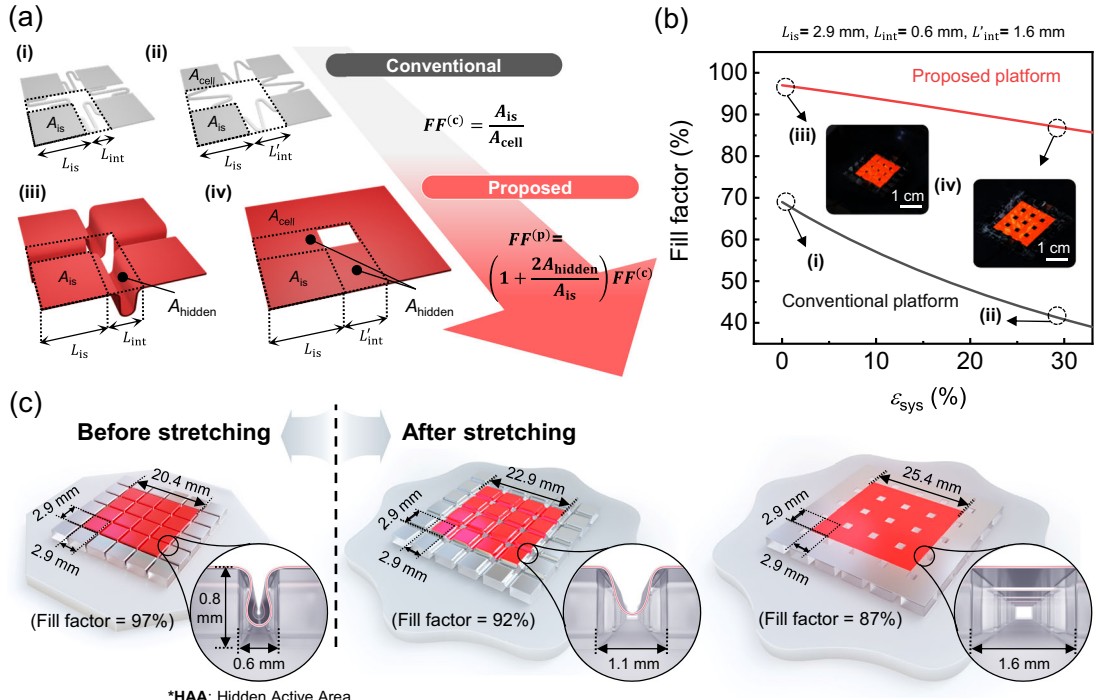

**Fig. 1 | An overview of the proposed high FF stretchable OLED with a hidden active area (HAA). a** Conceptual diagrams of (top) the conventional and (bottom) proposed platforms (left) before and (right) after stretching. **b** The calculated FF for both platforms according to the applied system strain ($\varepsilon_{sys}$). (Inset: Photographs of the realized high FF stretchable OLED driven at constant current of 1 mA before and after stretching in a tilted view) (i)-(iv) of **b** indicate the same states that are in (**a**).

The red line represents the proposed platform, while the black line represents the conventional platform's fill factor variation. **c** Schematic diagrams illustrating the gradual stretching process of the proposed stretchable OLEDs in three steps, along with dimensional information. The enlarged figure illustrates the detailed dimensions and appearance of the HAA.

$L_{is}$, and the length of the interconnectors before and after stretching as $L_{int}$ and $L_{int}'$, respectively. The system strain can then be determined using the equation $\varepsilon_{sys} = (L_{int}' - L_{int}) / (L_{is} + L_{int})$ commonly in both systems. Within a given unit cell area ($A_{cell}$) represented by the dotted line in (i) and (ii), the conventional platform has the active area ($A_{is}$) that remains the same before and after stretching. Therefore, the FF of the conventional platform ($FF^{(c)}$) is expressed by $A_{is}/A_{cell}$, and it usually degrades significantly due to the increased $A_{cell}$ when stretched. On the other hand, the proposed platform has additional two HAA areas ($2A_{hidden}$) per unit cell, resulting in an FF ($FF^{(p)}$) of $(A_{is} + 2A_{hidden})/A_{cell}$, which is $(1 + 2A_{hidden}/A_{is})$ times $FF^{(c)}$. By using these equations and following the calculation procedure described in Supplementary Note 1, the post-stretch FF of each platform can be expressed by Eqs. (1) and (2), by using $\varepsilon_{sys}$ and the initial FF corresponding to each of the cases $\left(FF_0^{(c)}, FF_0^{(p)}\right)$.

$$FF^{(c)} = \frac{FF_0^{(c)}}{\left(\varepsilon_{sys} + 1\right)^2} \approx \left(1 - 2\varepsilon_{sys}\right) FF_0^{(c)} \tag{1}$$

$$FF^{(p)} = \frac{FF_0^{(p)}}{\left(\varepsilon_{sys} + 1\right)^2} \left[1 + 2\varepsilon_{sys}\left(\frac{1 - \sqrt{1 - FF_0^{(p)}}}{FF_0^{(p)}}\right)\right] \approx (1 - 2\varepsilon_{sys}) FF_0^{(p)} + 2\varepsilon_{sys}\left(1 - \sqrt{1 - FF_0^{(p)}}\right) \tag{2}$$

where the last parts of the respective equations correspond to the approximated case for $\varepsilon_{sys} \ll 1$ in both Eqs. (1) and (2). As shown in Fig. 1b, $FF^{(c)}$ is decreased rapidly in proportion to $(1 + \varepsilon_{sys})^{-2}$, following Eq. 1. On the other hand, $FF^{(p)}$, given by Eq. (2), the decrement proportional to $(1 + \varepsilon_{sys})^{-2}$ is compensated by the factor of $2\varepsilon_{sys}\left(1 - \sqrt{1 - FF_0^{(p)}}\right)/FF_0^{(p)}$. Consequently, $FF^{(p)}$ is decreased at a much slower rate with respect to $\varepsilon_{sys}$ than $FF^{(c)}$, is always $> FF^{(c)}$ at a given $\varepsilon_{sys}$, and the improvement in FF by using the proposed HAA-based platform becomes larger as $\varepsilon_{sys}$ increases. For example, the FF of the proposed HAA-based platform exhibits an initial FF of 97% and a post-stretch FF of 87% at $\varepsilon_{sys} = 30\%$, while they are limited to 69% and 42%, respectively, for the conventional platform shown in Fig. 1a. The high initial and post-stretch FF of the proposed system is further verified by the images of the actual operating devices shown in insets (iii) and (iv) of Fig. 1b.

The schematic diagrams depicted in Fig. 1c represent three different stages corresponding to the initial (unstretched), half-stretched, and fully stretched states of the proposed stretchable OLED device with the HAA, illustrating how it can maintain a high FF for both initial and stretched states. In the initial state, the side length of the square island ($L_{is}$) and the width of the inter-island region between adjacent islands ($L_{int}$) are 2.9 mm and 0.6 mm, respectively. With the height of the side walls ($t_{is}$) set to 0.8 mm, the HAA region in this state is completely concealed in the inter-island region. Furthermore, we designed $L_{int}$ to be as small as possible to maximize the initial FF, which is near unity. This design choice, with a 0.6 mm spacing, prevents the HAA from touching the bottom surface of the inter-island region in the initial, non-stretched state, and ensures that different parts of the HAA do not adhere to each other. As the device is stretched, the island retains its original dimensions, but the HAA begins to rise toward the top, revealing its appearance gradually. When the device is fully stretched, the inter-island region is occupied by the flattened HAA region (1.6 mm wide), which enables non-disconnected light output in the inter-island region, which would otherwise be occupied with non-lit serpentine interconnectors. The reduction in the post-stretch FF compared to the initial FF in this case is due to the square void, which has an area of $A_{void} = (L_{int}')^2 = 1.6 \text{ mm} \times 1.6 \text{ mm}$, present in each cell. (A detailed explanation regarding the criteria set for each dimension can be referenced in Supplementary Note 1).

While the working principle might seem straightforward at first, its experimental realization presents challenges that require a carefully curated strategy. The first hurdle is to merge ultrathin OLEDs onto a rigid island array in a precise and mechanically reliable manner. Schematic diagrams presented in Fig. 2 summarize the step-by-step fabrication process for the proposed stretchable OLED with an HAA

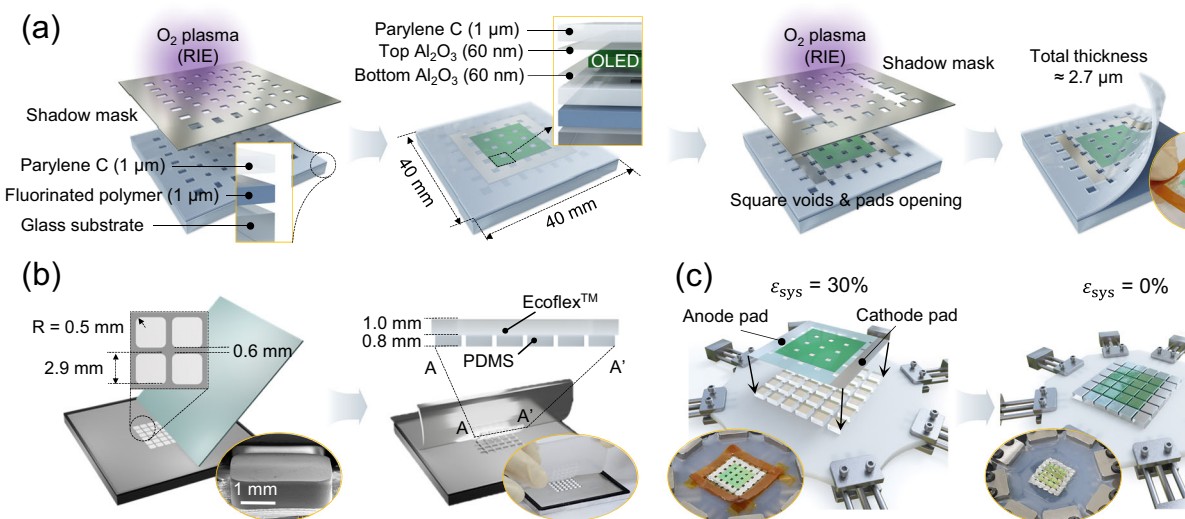

**Fig. 2 | Schematic diagrams illustrating the fabrication processes of the stretchable OLED in three steps. a** The fabrication process of the square void patterned Parylene C sandwiched OLED. From left to right, it illustrates the bottom Parylene C patterning, OLED deposition, top Parylene C patterning, and the ultrathin OLED peeling off process. (Inset: A photograph of the free-standing ultrathin OLED). **b** The process includes the fabrication of the patterned hybrid elastomer using doctor blading. (Inset: A SEM image of the patterned PDMS island and a photograph of the patterned hybrid elastomer). **c** The alignment and adhesion process of the Parylene C sandwiched OLED and patterned hybrid elastomer using a quadaxial stage. (Inset: Photographs of the transferred ultrathin OLED onto the pre-stretched patterned hybrid elastomer and the stretchable OLED after releasing the pre-strain).

structure in three main steps: (i) fabrication of ultrathin OLED foils based on Parylene C / OLED / Parylene C sandwich architecture having a pattern of the square void array (Fig. 2a), in which Parylene C films are 1 μm thick; (ii) preparation of a PDMS island arrays on an Ecoflex™ layer (Fig. 2b), the latter of which serves as a stress-relieving base layer that also functions as a layer that determines the overall stretchability of the whole system; and (iii) assembly of a patterned ultrathin OLED foil with the PDMS island arrays stretched on a quadaxial stage at a target maximum $\varepsilon_{sys}$. (Fig. 2c).

In the part (i), the Parylene C layer was chosen for its excellent hygroscopicity and ductility[43,44] as well as its mild and solution-free fabrication process, which is advantageous when being deposited on top of OLEDs. To facilitate easy detachment of the ultrathin OLED foil from a carrier glass, a 1 μm thick fluorinated polymer layer was spin-coated on a 40 mm by 40 mm glass substrate before preparing the bottom Parylene C layer. The formed Parylene C layer was patterned via reactive ion etching (RIE) with the assistance of a shadow mask to define an array of the aforementioned square voids. These square voids, spaced apart by $L_{is}$ (2.9 mm) from each other, serve a crucial role as open spaces that prevent any undesired irregular creasing when the devices transition from the stretched to the unstretched state. Although these holes are not visible in the unstretched state, they become evident once the device is stretched, leading to a reduced FF. Nevertheless, the dimensions of the voids are sufficiently small to maintain the high FF, reaching up to 87%, even in the fully stretched state. After the Parylene C patterning, a 60 nm-thick bottom $Al_2O_3$ film was deposited via atomic layer deposition (ALD) to form a gas barrier layer. Subsequently, OLED layers were formed via thermal evaporation in the top-emission configuration as follows: Al (100 nm) / {HAT-CN (10 nm) / NPB (60 nm)} 4 / TcTa (5 nm) / CBP: Ir(MDQ)$_2$acac (20 nm, 8 wt%) / BmPyPB (50 nm) / LiF (1 nm) / Al (1 nm) / Ag (20 nm) / NPB (60 nm), where HAT-CN, NPB, TcTa, CBP, Ir(MDQ)$_2$acac, and BmPyPB refer to 1,4,5,8,9,11-hexaazatriphenylenehexacarbonitrile, N,N'-Di(1-naphthyl)-N,N'-diphenyl benzidine, tris(4-carbazoyl-9-ylphenyl)amine, 4,4'-Bis(N-carbazolyl)−1,1'-biphenyl, bis(2-methyldibenzo[f,h]quinoxaline) (acetylacetonate) iridium(III), and 1,3-bis[3,5-di(pyridine-3-yl) phenyl]benzene, respectively. (Supplementary Fig. 1) The four pairs of HAT-CN (10 nm) / NPB (60 nm) were deposited as hole injection layers, which are thick enough to minimize electrical shorts and improve the reliability of the device[45,46]. Subsequently, a 60 nm-thick top $Al_2O_3$ and 1 μm-thick top Parylene C layer were further deposited. Pad opening for electrical contacts and top Parylene C patterning were carried out via RIE using a second shadow mask. Finally, the Parylene C sandwiched OLED with a total thickness of ca. 2.7 μm was gently peeled off from the carrier substrate, resulting in a free-standing structure. (Detailed photographs of each process step are shown in Supplementary Fig. 2).

Figure 2b depicts the fabrication process for the part (ii), which forms a patterned hybrid elastomer consisting of island arrays and a base. Initially, PDMS was poured onto an aluminum mold. Subsequently, the edge of a glass plate was used to doctor-blade away the excess PDMS, leaving behind the 6 × 6 island patterns. To ensure alignment accuracy during the fabrication process, the corner portion of the islands were designed to be rounded with the radius of 0.5 mm. (See the inset SEM image for the example of a PDMS island made in this way) Next, Ecoflex™ was poured to form a 1 mm thick layer and then cured. Afterward, this patterned hybrid elastomer, consisting of an Ecoflex™ base layer and a 6 × 6 arrays of PDMS on it, was separated from the Al mold. At this point, the free-standing ultrathin OLED foil described in part (i) was transferred and bonded onto the patterned hybrid elastomer. Firstly, the patterned hybrid elastomer was pre-stretched in four directions, including both orthogonal and diagonal directions, using a quadaxial stage. Then, an adhesive was applied onto the top of the island patterns. The ultrathin OLED foil was carefully aligned and bonded to the islands such that each of the square voids be

in between adjacent islands and bonded thereon. The quadaxial stretching implemented in the bonding process was vital for the proper mechanical operation of the proposed device; this will be further discussed in the following section. After the pre-strain was released, the whole unit was separated from the stage, resulting in a standalone stretchable OLED. (Detailed photographs of each process step are shown in the insets of Fig. 2b, c as well as in Supplementary Fig. 3).

**Mechanical simulation for integration via quadaxial stretching and design of patterned hybrid elastomer**

Figure 3a presents an enlarged schematic diagram illustrating the alignment of the proposed ultrathin OLED with respect to the island arrays of the patterned hybrid elastomer. Note that the integration is done under the maximum stretch condition with the ultrathin OLED made flat. To make the proposed plan work, the OLED should be aligned such that the corners of a square void may fall on the respective corners of the adjacent four islands. As the edge of the OLED has a straight outer contour (indicated by the blue dashed line in the figure) and the square void arrays are arranged periodically in a direction parallel to the blue dashed line, the island arrays have to be stretched straight at the uniform magnification ratio for proper alignment. To achieve this, we performed a mechanical simulation based on finite element method (FEM), which revealed that a quadaxial stretching method[42] is essential to allow for more precise control over the arrangement of the island arrays than the two widely used types of biaxial stretching: orthogonal[47,48] and diagonal stretching[34], as shown in Fig. 3b. In this simulation, we applied orthogonal strain ($\varepsilon_{ortho}$) and diagonal strain ($\varepsilon_{diag}$) to the island arrays at the same distance from the center of the elastomer for all three methods. (Detailed boundary conditions and assumptions are noted in the Methods section and Supplementary Fig. 4) To address the alignment accuracy in a more quantitative manner, we defined the displacement of an island with respect to a straight line as $y$ and aimed to minimize it. In the biaxial stretching methods, the strain along the orthogonal directions were set to be equal to ensure symmetry. ($\varepsilon_{ortho} = 30\%$, $\varepsilon_{diag} = 30\%$ for each biaxial stretching method) Nevertheless, the biaxial stretching turned out to result in pincushion deformation, leading to non-zero $y$ values that were non-negligible. In contrast, quadaxial stretching proved advantageous as the values of $\varepsilon_{ortho}$ and $\varepsilon_{diag}$ could be adjusted independently to ensure that all the islands aligned as closely as possible to the straight line.

To quantitatively analyze the impact of $\varepsilon_{diag}$ on the island alignment, we simulated $\Delta y$ as a function of the position $x$ for ($\varepsilon_{diag}/\varepsilon_{ortho}$) varying from 0 to 1, with $\varepsilon_{ortho}$ fixed at 30%. (Fig. 3c) As the relative portion of $\varepsilon_{diag}$ increases, $\Delta y$ values of the islands at both of the side edges gradually increase from negative (i.e. $|\Delta y|$ gets decreased). When $\varepsilon_{diag}$ is set at half of $\varepsilon_{ortho}$, all the islands in the bottom-most line form a straight line. As ($\varepsilon_{diag}/\varepsilon_{ortho}$) further increases, however, the $\Delta y$ value continues to increase and becomes positive. (Detailed simulation results and the positions of each island are indicated in Supplementary Fig. 5. The summary of the results is provided in Fig. 3d) Consequently, the quadaxial stretching exhibited a tight alignment accuracy under the condition of $\varepsilon_{diag}/\varepsilon_{ortho} = 0.5$, with $y$ ca. 0.01 mm between the islands and the straight line, being in clear contrast with the two biaxial cases that showed maximum $y$ values as large as 0.1 mm (orthogonal) and 0.3 mm (diagonal), respectively. (Detailed simulation results and the positions of each island are presented in Supplementary Fig. 6) Through this quantitative analysis, it becomes evident that $\varepsilon_{diag}$ plays a role in increasing the $y$ values for islands located at the vertices. Hence, we custom-designed and built a quadaxial stretching module, and used it to stretch the patterned-hybrid elastomer with ($\varepsilon_{diag}/\varepsilon_{ortho}$) set to 0.5, achieving distortion-less alignment between the ultrathin OLED and the rigid island array such that the corners of the square holes in the ultrathin OLED match the respective corners of the adjacent four islands.

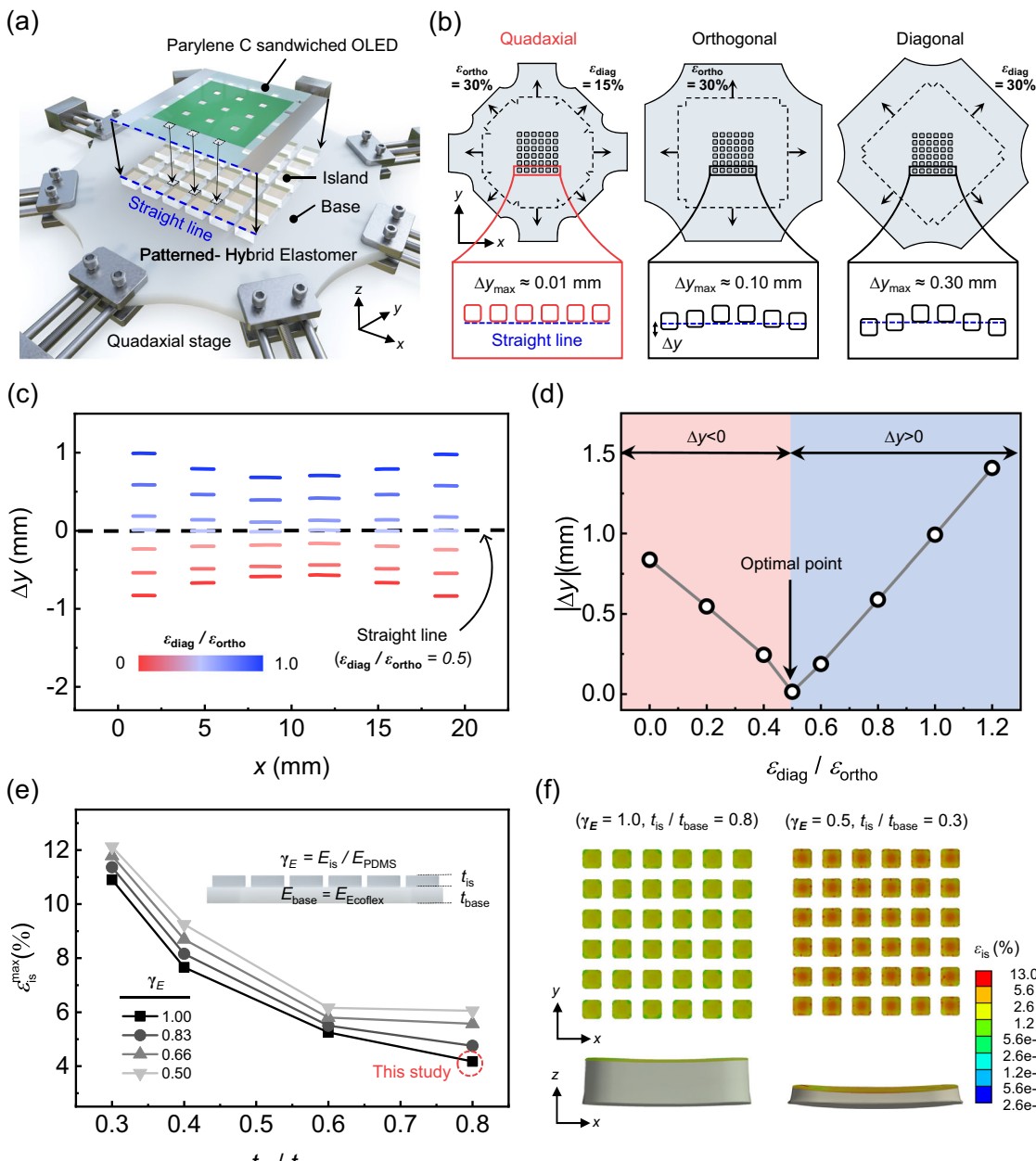

**Fig. 3 | Alignment using quadaxial stretching and FEM simulations for mechanical design of the patterned hybrid elastomer. a** An enlarged schematic diagram that illustrates the alignment of the Parylene C sandwiched OLED and the patterned hybrid elastomer. The dashed blue line represents a straight line that matches the ultrathin OLED pattern. **b** Schematic diagrams depicting three methods for stretching the patterned hybrid elastomer: quadaxial, orthogonal biaxial, and diagonal biaxial stretching. The dashed and solid lines represent the geometry before and after stretching, respectively. The magnified view shown right below compares the arrangement of islands stretched by each method with respect to that of a straight line. **c** The simulated $\Delta y$ values of the outermost six islands with respect to the straight line (depicted as a blue dashed line in **a**, **b**) vs. $\varepsilon_{diag}/\varepsilon_{ortho}$ ranging from 0 to 1. (The $\Delta y$ values become closest to zero when $\varepsilon_{diag} = 0.5\varepsilon_{ortho}$). **d** The simulated maximum $|\Delta y|$ values according to the $\varepsilon_{diag}/\varepsilon_{ortho}$. The red-highlighted area represents the region where $|\Delta y|$ is <0, while the blue-highlighted area represents the region where $|\Delta y|$ is >0. **e** The simulated $\varepsilon_{is}$ vs. various values of $t_{is}/t_{base}$ and $\gamma_E$ (= $E_{is}/E_{PDMS}$). (Inset: A detailed side view illustrating the simulation structure and the parameters). **f** The ANSYS simulation results present the equivalent strain distribution on the surface of a 6 × 6 island arrays and the deformation in the side profile of the islands for the case with the smallest $\varepsilon_{is}$ ($\gamma_E = 1.0$, $t_{is}/t_{base} = 0.8$) and the case with the largest $\varepsilon_{is}$ ($\gamma_E = 0.5$, $t_{is}/t_{base} = 0.3$).

The Young's modulus and thickness of the island ($E_{is}$, $t_{is}$) and the base ($E_{base}$, $t_{base}$) are additional parameters that require careful design consideration. Since the OLED is attached to the island, the bending rigidity of the island ($\propto E_{is}t_{is}^3$) should be sufficiently large when compared to that of the base to minimize the surface strain on the island ($\varepsilon_{is}$)[49]. Since the island and the base are directly attached to each other in the patterned hybrid elastomer, the system may be considered as a series connection of the two elements following Hooke's law[50]. For this reason, when a system strain is applied, the deformation occurs mainly

in the Ecoflex™ base, which has a relatively low Young's modulus, with the islands with the relatively high $E_{is}$. The simulation results shown in Fig. 3e confirm the beneficial role of high $E_{is}$ and $t_{is}$ on the reduction in $\varepsilon_{is}$. The simulation was conducted by (i) varying $\gamma_E = E_{is}/E_{PDMS}$, where $E_{PDMS}$ is Young's modulus of PDMS, from 0.5 to 1; and by (ii) varying the ratio of $t_{is}/t_{base}$ from 0.3 to 0.8 with $E_{base}$ and $t_{base}$ set at 25 kPa and 1 mm, respectively. It is observed that $\varepsilon_{is}$ decreases as $\gamma_E$ increases and $t_{is}/t_{base}$ ratio increases. Hence, a promising strategy to minimize $\varepsilon_{is}$ is to maximize the difference between $E_{is}$ and $E_{base}$ as long as $E_{base}$ is not

too low for easy handling. For this reason, we chose, as a base substrate material, Ecoflex™ 00–20, which has a lower Young's modulus (25 kPa) compared to other Ecoflex™ line-up. Additionally, we used, as an island material, PDMS mixed with a high proportion of curing agent (the PDMS base-to-curing agent ratio of 6:1) to ensure high $E_{PDMS}$, which turned out to be as high as 1.2 MPa. Upon examining the positive effect of $t_{is}/t_{base}$ ratio on reduction of $\varepsilon_{is}$, one might consider that maximizing $t_{is}$ can also be a promising strategy to minimize $\varepsilon_{is}$. However, an excessive increase in $t_{is}$ could result in a bulky system and poses limitation in detaching the island from the mold, as seen in the second image of Fig. 2b. Accordingly, we designed $t_{is}$ to be the highest among dimensions that allow for successful detachment from the mold, which was found to be 0.8 mm. The FEM simulation results shown in Fig. 3f further confirms the beneficial effect of high $t_{is}$ and $E_{is}$ on $\varepsilon_{is}$; the case with $\gamma_E$ = 1.0, $t_{is}/t_{base}$ = 0.8 exhibits a lower equivalent strain distribution on the surface of the 6 × 6 island arrays compared to the case with $\gamma_E$ = 0.5, $t_{is}/t_{base}$ = 0.3. Closer examination of the side view further revealed that the case with $\gamma_E$ = 1.0, $t_{is}/t_{base}$ = 0.8 exhibits less deformation in the islands compared to the case with $\gamma_E$ = 0.5, $t_{is}/t_{base}$ = 0.3. In short, considering these simulation results and fabrication feasibility, we have opted for the configuration characterized by $\gamma_E$ = 1.0, $t_{is}/t_{base}$ = 0.8.

## Strain analysis of mechanically brittle layers in HAA

When the ultrathin OLED is affixed to the pre-designed island arrays under full stretch and subsequently retracts to its unstretched state, HAA folds inwards, resulting in a curved cross-section in the inter-island region, as depicted in Fig. 4a. (It matches well with the inset SEM image) The geometric structure of HAA belongs to mode IV (self-contact buckling and partial delamination) among the deformation modes of compressed films adhered to a rigid substrate[51], and so it would be inaccurate to analyze it by approximating its profile as sinusoidal[52,53]. In this respect, we analyzed the equivalent strain distribution of the HAA using the finite element method (the structural mechanics module, COMSOL™ Multiphysics), and further details including boundary conditions and assumptions are noted in the Methods section and Supplementary Fig. 7. Special attention was given to the strain on the bottom and top $Al_2O_3$ layers as they have the smallest crack onset strain (COS) among all the layers involved. Note that, in this analysis, the points of interest for the strain value are A (A'), which is a point in contact with the edge of the island, B (B'), which corresponds to the inflection point of the curve, and C, where the radius of curvature is the smallest.

The strain at the bottom and top $Al_2O_3$ layers, estimated by varying the thickness of the Parylene C ($t_{pary}$) from 1 μm – 9 μm, are presented in Fig. 4b, c, respectively. Due to the unique structure of the HAA, where both ends are fixed to two adjacent islands and buckling occurs in between, a significant strain concentration effect occurs at point A (A'), resulting in the largest strain thereon. This strain concentration occurs due to the sharp corner that can be found at point A (A'). The level of strain concentration at the corner varies depending on the stiffness of a given structure, which is determined by its geometric parameters[54]. The simulation results indicate that the strain concentration effect at the point A (A') becomes more severe as $t_{pary}$ increases, because higher $t_{pary}$ yields a structure with higher stiffness. Upon closer examination, the values of strain tensors in the x, z, and

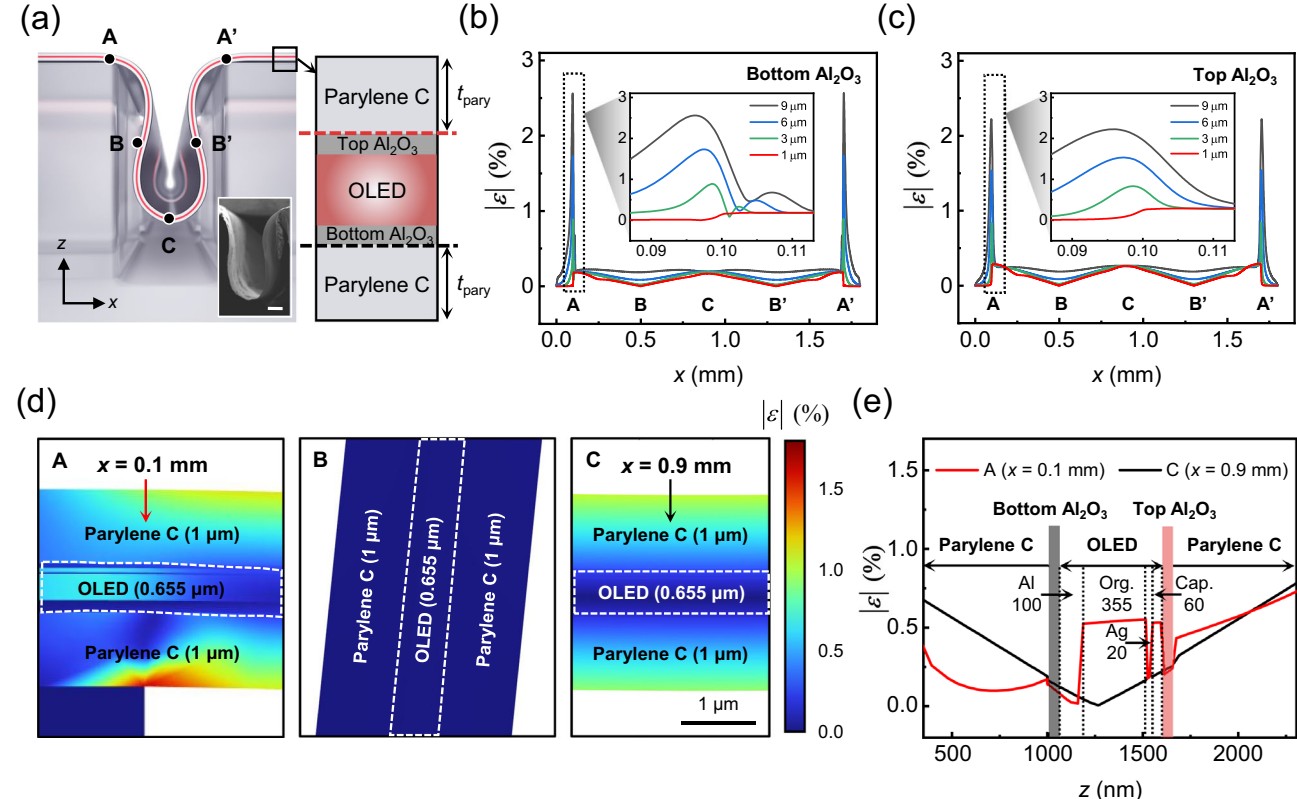

**Fig. 4 | FEM simulations of Parylene C sandwiched OLED with a HAA structure. a** A schematic diagram of the proposed HAA structure and the candidates (**A**, **B**, **C**, **A'**, **B'**) for strain concentration. In the layer structure on the right, the red dashed line indicates the point where the strain of the top $Al_2O_3$ is extracted, and the black dashed line indicates the point where the strain of the bottom $Al_2O_3$ is extracted. (Inset: A SEM image of the HAA is shown, with a scale bar indicating 100 μm) The simulated equivalent strain of (**b**) bottom $Al_2O_3$ and (**c**) top $Al_2O_3$ are depicted along the x-axis. (Inset: Magnified graphs of the equivalent strain concentration near part A (x = 0.1 mm)). **d** Enlarged views highlight the three parts (**A**, **B**, **C**) which indicate the equivalent strains in those areas. **e** The simulated equivalent strain along the z-axis is presented at part A (x = 0.1 mm) and part C (x = 0.9 mm). The numerical values displayed within the regions of each OLED layer denote the thickness of individual layers constituting the OLED, shown in units of nanometers (nm).

shear directions at the point A (A') consistently increase with $t_{pary}$. (Supplementary Fig. 8) Specifically, the dominant factors contributing to the overall increase in strain with thicker Parylene C layers were identified as the increase in $x$-axial strain tensor and shear strain tensor. This is because the HAA structure, originally expanded in $\pm x$-axis direction, is compressed in $\mp x$-axis direction, causing a restoring force that tends to increase strain tensor elements along $x$- and shear direction. To mitigate the adverse effects of the strain concentration at point A (A'), both the bottom and top Parylene C layers were designed to be very thin (~1 µm thick). As shown in the inset of Fig. 4b, c, the maximum strain on bottom and top $Al_2O_3$ in this case was maintained to be <0.3%, which is below COS[55].

Based on these insights, we designed the top-emission OLEDs sandwiched by 1 µm-thick Parylene C layers on their bottom and top. Figure 4d presents the zoomed-in view of the strain distribution across the whole layers composing the OLED including the Parylene C layers at A, B, and C. In the case of A ($x = 0.1$ mm), a significant strain is observed in the bottom Parylene C layer due to the aforementioned strain concentration effect; in the case of B, almost no strain is applied as it is the inflection point; and in the case of C ($x = 0.9$ mm), it is confirmed that the strain is smaller than A but not as small as B because C corresponds to the point with the minimal radius of curvature. Even with the relatively large strain applied to A and C, it was confirmed that the proposed structure ensures that the strain applied to the bottom and top $Al_2O_3$ in both A and C is below 0.3%. As shown in Fig. 4e, the strain distribution at point C exhibits a typical profile that is proportional to the distance from the neutral plane of each layer, without any specific peaks[55]. Upon closer examination of Fig. 4d and Fig. 4e, however, another strain concentration is observed at point A at the interface of Al/organic and organic/Ag/capping layers, attributed to the significant modulus mismatch between the layers[50]. Nevertheless, the strains on the anode (Al) and cathode (Ag) layers are still <1% and thus remain below their respective COS values, justifying the use of the proposed design in terms of reliable HAA operation.

## Performance of the proposed stretchable OLEDs under various mechanical deformations

Figure 5 illustrates the electrical-optical characteristics of the realized stretchable OLED under various mechanical deformations. As shown in Fig. 5a, it can be observed that the current density ($J$) and luminance ($L$) versus voltage ($V$) characteristics do not change significantly as the devices are biaxially stretched, at $\varepsilon_{sys}$ varied from 30% to 0% at 10% intervals in each orthogonal direction. ($\varepsilon_{sys} = \varepsilon_x = \varepsilon_y = 30\%$) (Additional experimental data and a detailed explanation regarding the difference in $J$-$V$-$L$ characteristics in the high voltage region are provided in Supplementary Fig. 9) Furthermore, the normalized current efficiency ($\eta_{CE}$) values at 100 cd m$^{-2}$ were shown to maintain their initial value (4.5 cd A$^{-1}$) up to 97% when the device was fully stretched. (Fig. 5b, Supplementary Fig. 10) The inset images also confirm that the OLED driven at constant current of 1 mA under $\varepsilon_{sys}$ of 0%, 10%, 20%, and 30% operated without any visual degradation. (Scale bar corresponds to 1 cm in length.) Additionally, the HAA parts rose to the surface as the device was stretched, turning the interconnector area, which would otherwise be optically inactive, into an active lighting region, thereby minimizing the FF loss induced by stretching. (Inset photographs in Fig. 5b).

Furthermore, apart from the degradation caused by the mechanical deformation of stretchable OLED, non-uniform brightness and color distribution, depending on the amount of $z$-axis deformation, are observed in the inset images of Fig. 5b. Specifically, the interconnector area appears brighter than the rigid island area when the device is not stretched. This phenomenon can be explained as light concentration, as depicted with ray diagrams in Supplementary Fig. 11a. In the initial state, rays emitted from the surface of the HAA undergoes multiple reflections within the structural elements of the

HAA, resulting in a light output through the narrow, confined space and thus making the effective luminance enhanced locally. If this non-uniformity should be avoided in target applications, matrix addressing may be introduced so that the light output in the HAA region may be adjusted separately depending on the strain level.

In addition, it is also observed that, in the non-stretched state, light from the HAA region exhibit a spectrum that is shifted from that observed in the central region of the rigid islands. This is because light from the HAA region tends to exhibit relatively high-angle emission with respect to the emitting surface. In Supplementary Fig. 11b, c, the angle-dependent variations in EL spectrum and CIE color coordinates ($x$, $y$) values measured with the planar, reference OLED are shown. The significant changes in the spectrum and CIE values are attributed to the viewing angle-dependent color shift characteristic of the 2$^{nd}$ cavity top emission OLED structure. This observation is consistent with the results of the CIE color coordinates ($x$, $y$) values measured for the proposed stretchable OLEDs, as shown in Supplementary Fig. 12; CIE $x$ changes from 0.71 (at island) to 0.58 (at HAA), while CIE $y$ changes from 0.29 (at island) to 0.39 (at HAA). Nevertheless, the viewing angle-dependent color shift can be suppressed to a significant degree through a careful microcavity design only with a slight sacrifice in the efficiency[56].

After confirming the mechanical stability at each strain level, the proposed stretchable OLED was tested for the mechanical reliability under repeated cycles of deformation using a home-made computer-controlled biaxial cyclic test equipment, as shown in the inset in Fig. 5d and Supplementary Fig. 13. With this set up, we measured $J$-$L$-$V$ characteristics under biaxial strain of 30% after 1, 10, 100, 1000 'full stretch and release' cycles. (Fig. 5c) Although $J$ and $L$ at a given voltage decreased slightly with the number of cycles, $\eta_{CE}$ remained almost intact, as shown in Fig. 5d. (See Supplementary Fig. 14 for the lifetime test, and Supplementary Movie 1 for actual operation) These experimental results verify that the proposed architecture, designed through simulations in Fig. 4, effectively reduces the strain concentration effect and ensures sufficiently low strain applied on both bottom and top $Al_2O_3$ layers for mechanically reliable operation. They demonstrate that the fabricated high FF stretchable OLED exhibits high mechanical reliability even under 30% biaxial strain and 1000 cycles of repeated operation. Furthermore, it was illustrated that the fabricated stretchable OLED operated well after being conformally attached to the surfaces of a spherical object (left) and a cylindrical object (right) with a radius of 1 cm (Fig. 5e), as well as to an ellipsoidal surface like a balloon that stretches in uneven manner (Fig. 5f and Supplementary Movie 2). It was further demonstrated that the device was well operable when it was attached to human body parts, such as the back of hands and elbows, or when it was stretched using hands in arbitrary biaxial directions (Supplementary Fig. 15). As the proposed stretchable OLEDs are designed primarily for biaxial stretching, uniaxial stretching could render the interconnections perpendicular to the stretching direction subject to a relatively large stress due to the substrate's Poisson's ratio. Our analysis indicates that the devices are still operable under uniaxial stretching with a substrate strain of up to 10%, although the value higher than that could make devices face geometrical limitation. (Supplementary Fig. 16 for further details) All these results illustrate that our high FF stretchable OLED can be applied to various curved surfaces and is ideal even for a wearable light source that can accommodate dynamic deformation induced by stretching, for instance, due to the physical movement, etc.

## A stretchable passive matrix OLED display without post-stretch resolution reduction

Utilizing HAA as hidden pixels enables the implementation of an advanced display that compensates for resolution decrease during stretching. Figure 6a, b represent conceptual diagrams of a conventional platform without hidden pixels and the proposed platform

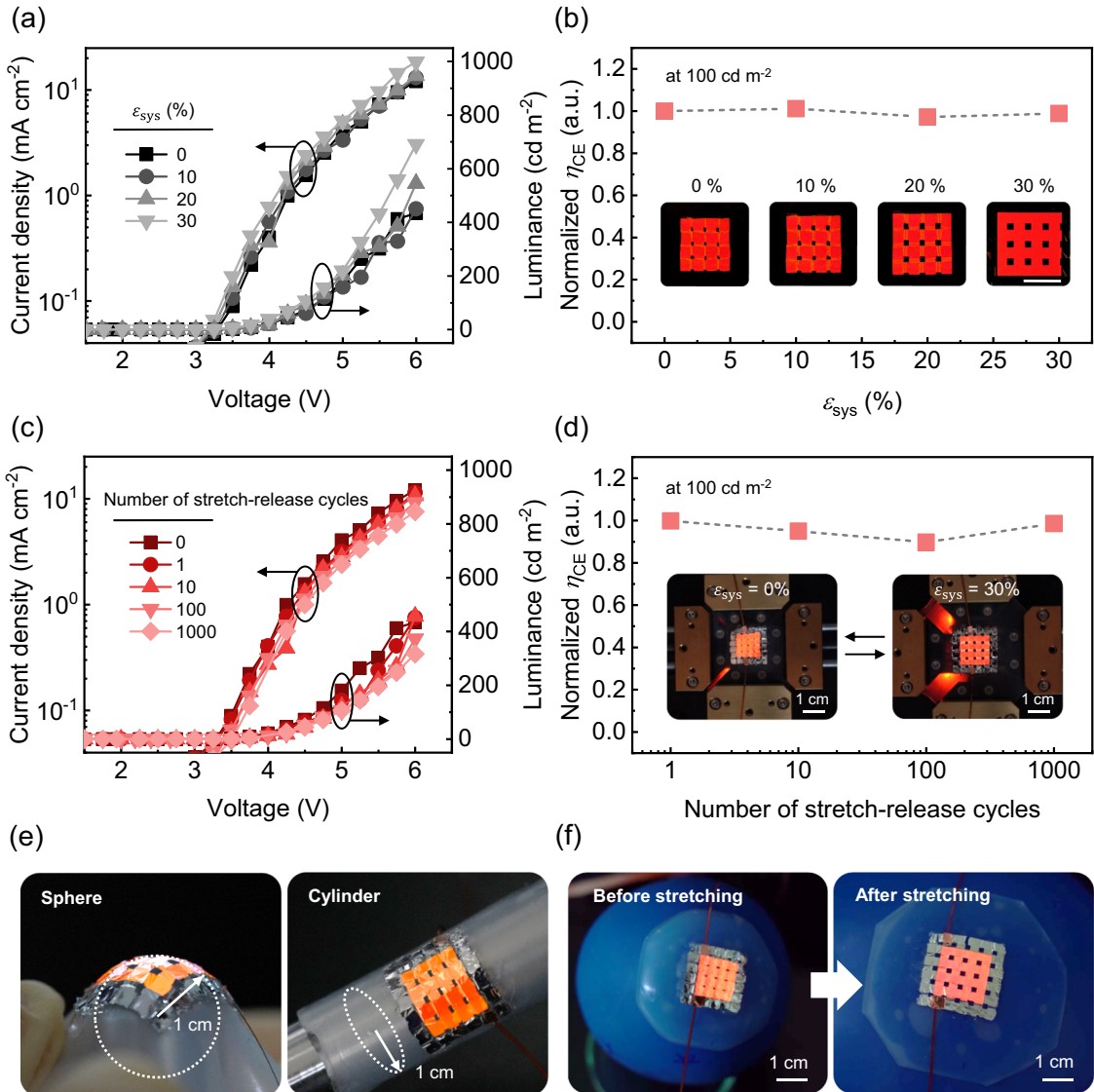

**Fig. 5 | The performance of the high FF stretchable OLEDs under various mechanical deformations. a** The current density and luminance characteristics versus driving voltage for the different biaxial system strain ($\varepsilon_{sys}$). **b** The normalized current efficiency ($\eta_{CE}$) at 100 cd m$^{-2}$ according to $\varepsilon_{sys}$. (Inset: Photographs of the stretchable OLED for $\varepsilon_{sys}$ ranging from 0% to 30% with 10% interval. A scale bar indicates 1 cm). **c** The current density and luminance characteristics versus driving voltage according to the numbers of stretch-release cycles at $\varepsilon_{sys}$ = 30%. **d** The

normalized current efficiency ($\eta_{CE}$) at 100 cd m$^{-2}$ according to the number of stretch-release cycles at $\varepsilon_{sys}$ = 30%. (Inset: Photographs of the operating stretchable OLED loaded in the biaxial cyclic test equipment with $\varepsilon_{sys}$ = 0% (left) and $\varepsilon_{sys}$ = 30% (right). **e** Photographs of stretchable OLEDs conformally attached to a sphere (left) and cylinder (right) with a radius of 1 cm. **f** Photographs of stretchable OLEDs conformally attached to a balloon before (left) and after (right) expansion. All of the operating OLEDs in photographs were driven with a constant current of 1 mA.

compensating for resolution decrease through hidden pixels, respectively. Building upon the proposed stretchable OLED platform with HAA, we further designed a passive matrix array system, in which 40 pixels can be operated individually in a 7 × 7 array structure, excluding the central 9 voids. Detailed information about the passive matrix (PM) stretchable OLED display is provided in Supplementary Fig. 17 to **21** and the Experimental section. To illustrate the limitation of conventional rigid-island-based stretchable displays, we first operated the proposed PM stretchable OLED with its hidden pixels turned off, mimicking conventional stretchable displays. In this case, not only the initial FF is very low at 18%, but also the post-stretch FF decreases to 11% when stretched with a 30% biaxial strain. Consequently, even in the initial state, the legibility of alphabets is rather poor due to its relatively large pixel-to-pixel distance (Fig. 6c and Supplementary Movie 3). When the display is stretched, the situation gets aggravated (Fig. 6d and Supplementary Movie 4). On the other hand, with the

proposed platform that incorporates hidden pixels, the initial FF increases significantly to 33%, and the post-stretch FF increases to 26%, compared to the conventional case. As a result, the legibility of alphabets is significantly improved due to its initial high pixel density (Fig. 6e and Supplementary Movie 5) which is kept well even after stretching (Fig. 6f and Supplementary Movie 6) thanks to the pixel-density compensation enabled by HAA. This clearly illustrates the immense benefits of the proposed platform in stretchable display applications. It is noteworthy that the fill factor of PM OLEDs that appear relatively low may not be regarded as an intrinsic limitation of the proposed methodology. If industrial-grade equipment such as a vision alignment system is used to secure a very tight alignment margin, it is plausible to achieve a fill factor that is quite comparable to that of the common-electrode type stretchable OLEDs, although it will still be limited to ensure line-to-line isolation for prevention of cross-talk.

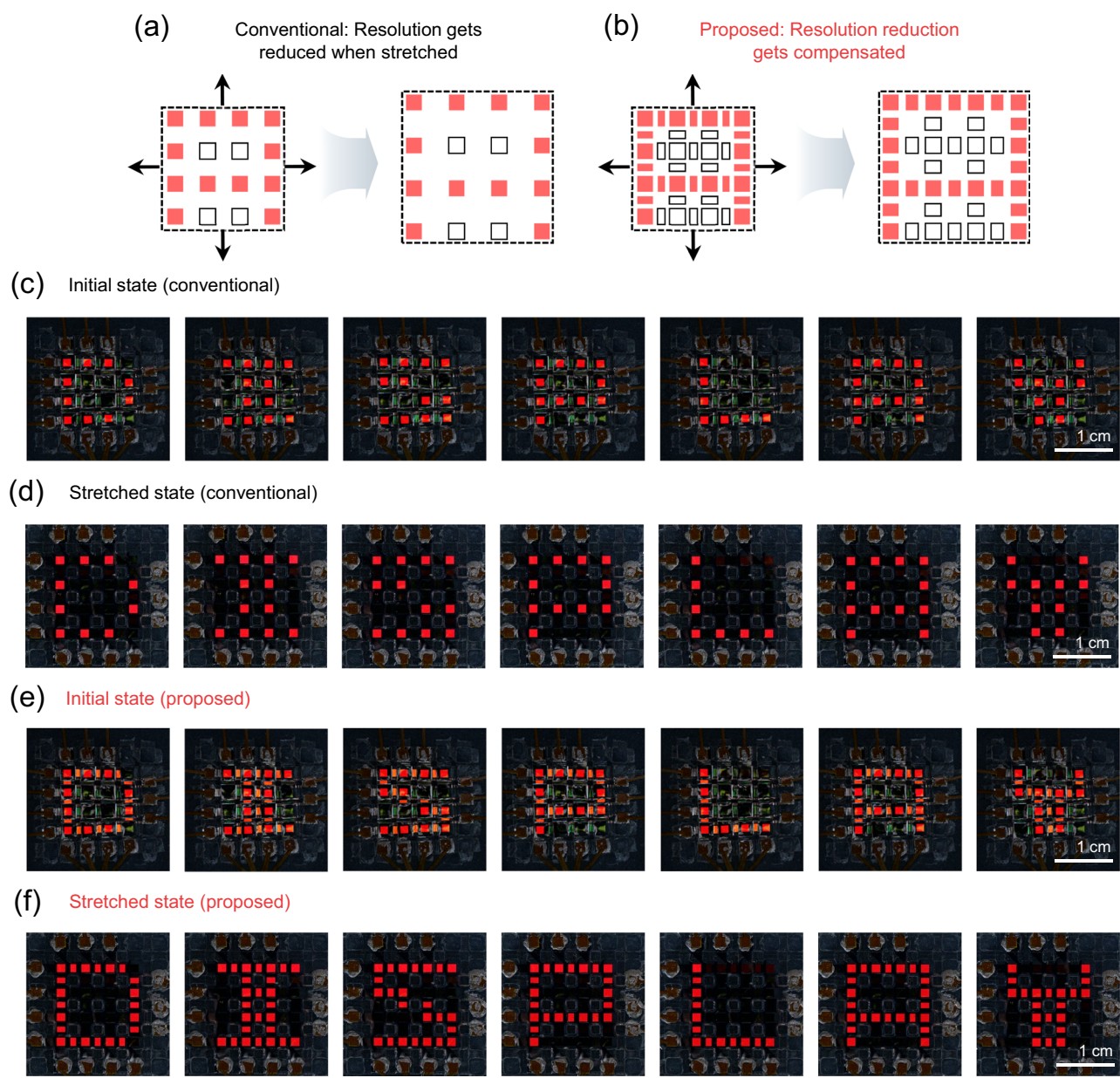

**Fig. 6 | The stretchable passive-matrix OLED (PM OLED) display with hidden pixels.** The concept diagrams depicting the initial and stretched states of (**a**) a conventional display and (**b**) the proposed display with hidden pixels. Photographs displaying the 'D', 'I', 'S', 'P', 'L', 'A', and 'Y' alphabets through PM stretchable OLEDs in (**c**) the initial state and (**d**) the stretched state without hidden pixels, as well as in (**e**) the initial state and (**f**) the stretched state with hidden pixels.

To apply this platform in advanced display applications, it is important to keep the compatibility with high pixel-per-inch (PPI) density and attain a high production yield. It should be noted that, although each island hosts only one pixel in the present demonstration, each island as well as the HAA region can host multiple pixels, respectively. This is possible because the device is first fabricated on a planar substrate and then transferred to an array of PDMS islands on an elastomer. In such a scheme, a higher level of pixel density for moderate-to-high resolution displays could be within reach. At the same time, the proposed scheme is regarded advantageous for higher production yield while the conventional stretchable LED displays often encountered low yields due to high failure rates when transferring multiple LED units[57,58]. It is also noteworthy that the present stretchable displays inherit the high-speed characteristics of OLEDs[59–62], which makes them readily capable of video-rate operation. (See Supplementary Movies 7 and 8 for 10- and 100 Hz operation of the proposed stretchable OLEDs in comparison with 10 Hz operation).

## Discussion

In summary, our research introduced the concept of the hidden active area (HAA) as a method to minimize the stretching-induced reduction in FF by combining ultrathin OLEDs and a 3D rigid island array structure in which the HAA is concealed in between adjacent islands in the initial non-stretched condition and gradually surfaces to the top when stretched. The HAA serves dual roles as (i) an alternative to stress-relieving interconnectors like serpentine interconnector, which maintains the main active area intact while enabling the stretching of the overall system; (ii) and as an auxiliary active area that gradually lifts up upon stretching, compensating for the FF loss inevitable in a stretchable system composed of rigid islands and serpentine connectors.

A fabrication process to realize high FF stretchable OLEDs containing the HAA involved aligning and bonding the patterned hybrid elastomers, composed of rigid high-modulus islands and a base low-modulus substrate, with ultrathin OLEDs sandwiched by top and bottom 1-μm-thick Parylene C films. We showed that it was important to

use quadaxial stretching to ensure precise alignment, as it offered superior control over strain distribution compared to the conventional biaxial stretching technique. FEM simulations assisted in optimizing materials and dimensions in stretchable OLEDs, thus minimizing strain on the constituent layers. Particular attention was paid to the HAA structure, which experienced bending-induced strain when the system retracted from the fully stretched state to the initial unstretched state. Simulation studies indicated that the point where the HAA was in contact with the edge of the rigid island experienced the highest strain. It was found that the proposed HAA design ensured the strain on all the layers, including $Al_2O_3$ and electrode layers, remained below their crack onset strain (COS) throughout the device. With this proposed concept and optimal design, the proposed stretchable OLEDs exhibited a near-unity initial FF, which was well-maintained with only a ca. 10% decrease, even under a 30% biaxial system strain. Note that this is a significant step-forward over traditional rigid island arrays with stress-relieving interconnectors, which could experience ca. 60% decrease for the same applied biaxial system strain. The current efficiency was also confirmed to be consistent within an acceptable range. In addition, the stretchable OLED was demonstrated to be useful for a variety of applications, including attachment to curved surfaces such as spheres, cylinders, ellipsoids and to movable human body parts for wearable applications that can accommodate dynamic joint motions, etc. Combined with their high FF for both initial and stretched states, the mechanical reliability of the proposed stretchable OLEDs paves the way for advanced displays and versatile lighting solutions that can conform to freeform surfaces. Furthermore, the HAA concept can be extended to other electronic components where high geometrical FFs are critical, such as image sensors and photovoltaic (PV) cells. This shows great promise for deformable and conformable optoelectronic devices, which are crucial for the successful deployment of future electronic systems.

## Methods

### Fabrication of patterned hybrid elastomer fabrication

To produce a mold for the patterned hybrid elastomer, a 70 mm by 70 mm aluminum plate with a thickness of 5 mm was utilized to engrave negative patterns of square islands. In order to promote easy detachment without residue of the patterned hybrid elastomer, which consists of Polydimethylsiloxane (PDMS) (Sylgard® 184) and Ecoflex™ (Smooth-on, Inc.), from the mold after fabrication, the surface of the aluminum mold was fluorinated through self-assembled monolayer (SAM) treatment using Trichloro(1H, 1H, 2H, 2H perfluorooctyl)silane (Sigma-Aldrich). To perform the SAM treatment, the aluminum mold was subjected to air plasma of 100 W for 10 min (CUTE™, Femto Science) to grow an oxide layer on the aluminum surface. Subsequently, the mold with the oxide surface was placed in a vacuum environment together with a small amount of the silane material in a desiccator for 30 min, and followed by an annealing at 150 °C for 10 min. Next, PDMS was prepared by mixing a base and a curing agent in 6:1 ratio and poured onto the mold. Excess PDMS outside the engraved island patterns in the mold was removed by doctor blading. Remaining PDMS inside the engraved patterns was slightly cured by resting for 10 min at room temperature. The mold was then filled with Ecoflex™ 00–20 and cured at 100 °C for 1 h. Finally, the cured patterned hybrid elastomer was gently peeled off from the mold.

### Fabrication of ultrathin organic light-emitting diodes (OLEDs)

Before fabricating an ultrathin OLED, we prepared a 40 mm by 40 mm carrier glass of 40 mm by 40 mm coated with a 1 μm-thick fluorinated polymer (Novec™ 1700 Electronic Grade Coating, 3M™) film on top. This was achieved by using spin-coating at 2000 rpm for 30 s, followed by annealing at 150 °C for 10 min. We then deposited a 1 μm-thick layer of bottom Parylene C onto this fluorinated polymer layer with a Parylene coater (OBT-PC300, OBANG TECHNOLOGY) under a base

pressure of $1.2 \times 10^{-2}$ torr. Subsequently, we patterned the bottom Parylene C layer to have an array of square voids using reactive ion etching (RIE) for 10 min with $O_2$ inflow of 25 sccm and RF power of 300 W. These square void patterns were defined by a shadow mask affixed on top of the bottom Parylene C layer during the RIE process. We then deposited a 60 nm-thick $Al_2O_3$ layer using atomic layer deposition (ALD) (LUCIDA™ D100, NCD) at a base pressure of $3.0 \times 10^{-1}$ torr in 70 °C for the bottom encapsulation layer. OLED layers were deposited onto the $Al_2O_3$ layer by vacuum thermal evaporation in a vacuum chamber at a pressure of $5 \times 10^{-7}$ torr. The top encapsulation layer of 60 nm-thick $Al_2O_3$ layer and a 1 μm-thick top Parylene C layer were deposited sequentially onto the OLED layers. The top Parylene C layer was patterned by the RIE process with a shadow mask to contain the square void array identical to that in the bottom parylene C layer and pad openings to contact the anode and cathode. Finally, we peeled the ultrathin OLED, with its Parylene C sandwiched structure, off the substrate wherein the interface between the fluorinated polymer and the bottom Parylene C layer served as a delamination interface.

### Combining the patterned hybrid elastomer and the ultrathin OLED

The patterned hybrid elastomer was pre-stretched using the quadaxial stretching module to align the island array position in the elastomer and those in the ultrathin OLED, as illustrated in Fig. 3 and Supplementary Fig. 3. To apply an adhesive material (DOWSIL™ SE 9186) uniformly onto the island arrays of the patterned hybrid elastomer, a glass substrate with a freshly spin-coated adhesive was briefly brought into contact with the patterned hybrid elastomer. As a result, the adhesive was transferred only to the top surface of the island arrays. The spin-coating process for the adhesive material onto a glass was performed at a speed of 4000 rpm for 30 s. The free-standing ultrathin OLED with the square void array was, then, attached to the patterned hybrid elastomer, ensuring alignment to match the positions of the island arrays, and left at room temperature for 1 h for curing of the adhesive between two components. Finally, the pre-strain was released, with a glass substrate placed on top to ensure downward buckling[38], thus resulting in the completed stretchable OLED having the proposed structure.

### FEM-based mechanical simulation for patterned hybrid elastomer

Static structure tools in ANSYS program was utilized to simulate the stretch behavior of the patterned hybrid elastomer. The stress-strain curves of PDMS (6:1) and Ecoflex™ 00–20 were measured using motorized force test equipment (ESM303 tensile tester, Mark-10), and used for realistic simulation. From the curve, the mechanical properties of Ecoflex™ 00–20 was properly fitted using Yeoh's 2nd order hyperelastic model[63]. The Young's modulus and Poisson's ratio of PDMS were set to 1.2 MPa and 0.49, respectively. The simulation results were presented as equivalent elastic strain values. In ANSYS™ program, the red-colored regions shown in Supplementary Fig. 4 were set as surfaces to be grabbed to initiate and maintain displacement. The full displacement was divided into 13 steps for simulation. No additional boundary conditions were applied to other surfaces, and it was assumed that there was no delamination between PDMS and Ecoflex™.

### FEM-based mechanical simulation for OLED layers on hidden active area (HAA)

To investigate the effect of Parylene C thickness on the strain distribution of individual OLED layers on HAA, the structural mechanics module of the COMSOL™ Multiphysics program was employed. This approach accommodates mesh settings with very large aspect ratio, and this is highly advantageous in the simulation involving thin-film structures such as OLED. In this simulation, the Young's modulus

values of all organic layers in the OLED structure was set to be 2.75 GPa, and the LiF hole injection layer and the Al seed layer were excluded because of their ignorable thickness of 1 nm. Additionally, the Young's modulus values of the remaining layers, Parylene C, $Al_2O_3$, Al, and Ag, were respectively set to 2.15 GPa, 134 GPa, 140 GPa, and 70 GPa[55]. To replicate the HAA structure in the simulation which has downward buckling within two PDMS islands, one PDMS island was gradually moved closer to the other PDMS island from their initial distance of 1.6 mm–0.6 mm in final with ten sub-steps as shown in Supplementary Fig. 7. Pre-described displacement boundary conditions of +500 μm on the left PDMS surface and -500 μm on the right PDMS surface were applied, while the remaining surfaces were assumed to be free. It was assumed that there was no delamination between PDMS and the thin-film OLED.

## Electro-optical performance evaluation of stretchable OLED under mechanical deformation

The electro-optical performance of the stretchable OLED device, encompassing both current density ($J$) and luminance ($L$), was assessed using an imaging spectrophotometer (CS 2000, KONICA MINOLTA, Inc.) in conjunction with a programmable source meter (Keithley 2400) under atmospheric conditions. The current efficiency ($\eta_{CE}$) was determined based on the measured $J$ and $L$. The CIE color coordinates across the entire surface of the OLED were evaluated using a 2D color analyzer (CA 2000, KONICA MINOLTA, Inc.). And the angular electroluminance (EL) spectra were measured with a customized measurement system consisting of a fiber optic spectrometer (BW_UVNb, StellarNet) held on a motorized goniometer (PRM1/MZ8, Thorlabs) for angle-resolved measurements. To evaluate the stretchable OLED under mechanical deformation, we employed a custom-made biaxial cyclic test apparatus based on two motorized linear translators (T-LSR075D, Zaber Technologies, Inc.). The stretch-release cyclic test was carried out with a biaxial system strain of 28.6%, measuring the device characteristics the CS 2000 after 1, 10, 100, and 1000 cycles. The motor operation speed was set to 200 mm min⁻¹ with a delay time of 0.1 s. The lifetime test was conducted for the reference OLED as well as for the stretchable OLED before and after the cyclic test. A calibrated photodiode (FDS 100-cal, Thorlabs) was used to measure the relative change in the brightness level as a function of operation time. The reference OLED was fabricated on a 4 cm × 4 cm glass substrate, and all three types of devices were operated under constant current driving conditions corresponding to the initial brightness of 100 cd m⁻².

## Fabrication and demonstration of passive matrix stretchable OLEDs

As an example of display application, 7 × 7 array of PM stretchable OLEDs was realized. The bottom and top encapsulation layers, as well as the organic layers of the PM OLEDs, remained consistent with the previously fabricated OLEDs. We designed 7 data lines and scan line electrodes, sacrificing a part of the FF, although kept minimal. (Supplementary Fig. 17) Given the small pixel spacing in the micrometer scale, we employed a self-aligning method, sequentially depositing each layer using guide masks and "puzzle masks" with magnets[2] to secure an alignment margin as small as -100 μm. (Supplementary Fig. 18) To detour the central 9 holes, we designed the spacing between the anodes to be 250 μm and between the cathodes to be 200 μm, resulting in a densely wired structure. Consequently, the emission area was 1.5 mm × 1.5 mm for the island, and 1.5 mm × 1.0 mm for the HAA. (Supplementary Fig. 19)

As previously discussed in Fig. 4, designing Parylene C with a thickness of 1 μm significantly reduces the strain concentration effect, ensuring mechanical stability. However, an excessively thin Parylene C thickness might lead to drawbacks in the manufacturing process, such as substrate tearing or challenges in the transfer process, causing lower yields. Unlike the previous full-area device, in the case of the PM OLED, the active area does not reside at the hinge part (Fig. 4, point A (A′)). Consequently, there is more flexibility in the design of Parylene C thickness as long as the strain imposed on the electrode layers remains below the COS. Therefore, to leverage the advantage of enhancing the process yield, a 3 μm-thick Parylene C layer was employed. At this thickness, the strain applied to points A (A′) and C (C′) remains below 1%, allowing for a sufficiently stable design to protect the electrodes (Fig. 4b, c). Finally, a customized FPCB was connected to the OLED pad using Ag paste and was driven through an Arduino (Arduino®, Arduino mega 2560). To drive seven alphabets ('D', 'I', 'S', 'P', 'L', 'A', 'Y') using 4 × 4 and 7 × 7 pixels, as shown in Supplementary Fig. 20, the voltages applied to each line are illustrated in Supplementary Fig. 21 (5 V for high, 0 V for low).

## Data availability

The authors declare that the main data supporting the findings of this study are available within the article and its Supplementary Information files. The source data underlying Figs. 1b, 3c,–e, 4b, c, e, 5a,–d, Supplementary Fig. 8a, b, c, 9, 10, 11b, c, 14c, 16c are provided in the Source Data files, and extra data are available from the corresponding author on request. Source data are provided with this paper.

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

## Acknowledgements

D.G.L., S.B.K., T.H.K., D.H.C., J.H.S., W.C.L., J.H.K., S.I.H., H.M., and S.Y. would like to acknowledge the financial support from the Engineering Research Center of Excellence (ERC) Program supported by the National Research Foundation (NRF), Korean Ministry of Science and ICT (MSIT) (Grant No. NRF-2017R1A5A1014708). H.S.C. and J.H.Y. would like to acknowledge the financial support from the Electronics and Tele-communications Research Institute (ETRI) grant funded by the Korea government. (21ZB1200, The Development of the Technologies for ICT Materials, Components and Equipment).

## Author contributions

S.Y. and H.M. conceived the idea of the stretchable OLEDs with hidden active area, and D.G.L., S.B.K., and T.H.K. designed the associated experiments. D.G.L., H.S.C., J.-H.Y., J.H.S., and W.C.L. fabricated and tested stretchable OLEDs with Parylene C sandwiched OLED and patterned hybrid elastomer. D.G.L, J.H.K., and S.B.K. conducted FEM-based mechanical simulations using the structural mechanics module of COMSOL Multiphysics and ANSYS. D.G.L., D.H.C., and S.I.H conducted the demonstration and measurement of the fabricated devices. D.G.L. analyzed all of the data, and D.G.L. and S.Y. wrote the manuscript and coordinated all the experiments. All authors read and discussed the results of the manuscript. S.Y. and H.M. contributed equally as corresponding authors.

## Competing interests

The authors declare no competing interests.
