## [Peer Review File · Nature Communications]

Stretchable OLEDs based on a hidden active area for high fill factor and resolution compensationREVIEWER COMMENTS

Reviewer #1 (Remarks to the Author):

This article introduces a novel design for biaxially stretchable OLEDs, aimed at enhancing the fill factor by employing concealed active areas as stretchable interconnectors. A quadaxial pre-stretching strategy was implemented on a substrate with meticulously controlled modulus, resulting in an undistorted island arrangement. Furthermore, OLED configuration simulations were conducted to mitigate strain on the device. While this study holds promise for high fill factor stretchable OLEDs, the advancements made, in comparison to existing literature, appear to be somewhat limited. As a result, concerns may arise regarding its suitability for publication in Nature Communications.

Comment 1: To distinguish this work from previously reported findings, it is essential to include an active matrix or passive matrix array demonstration. The current setup lacks a substantial differentiation from previously documented wrinkled or buckled biaxially stretchable LEDs, bearing a striking resemblance to instances such as "Nat. Commun. 2013, 7, 11573." and "Nano Lett. 2020, 20, 1526." Hence, the authors are advised to exhibit an array that showcases individual pixel control capabilities.

Comment 2: The descriptions of the simulation results in figures 3 and 4 appear overly simplistic. It is imperative that the authors engage in a comprehensive scientific discussion of the results, moving beyond mere reporting.

Comment 3: In the present configuration where the island remains entirely affixed to the substrate, there is a potential for uniaxial stretching to subject the interconnections perpendicular to the stretching direction to greater stress due to the substrate's Poisson's ratio. The reviewer raises a concern regarding whether this could result in device failure, or if the quadaxial pre-stretching approach is effective in averting such a situation.

Comment 4: Numerous errors have been identified in the references, necessitating thorough proofreading and correction.

Reviewer #2 (Remarks to the Author):

This paper reports 3-dimensional structure containing hidden active area by combining ultrathin OLED sheet and elastic pillar array in order to solve technical issues of resolution reduction and fill factor lowering under deformation, for stretchable light-emitting devices based on an island-bridge structure. The fabricated devices showed high fill factors of 97 and 87% before and after 30% stretching deformation and excellent reliability after 1000-time deformation. Authors well demonstrated stable stretchable devices on various curved and body surfaces and logical analysis on fabrication process and mechanical stability of the proposed structures using FEM simulation. After provide some of the issues described below, the paper is good for publication in Nature Communications.

1. Differentiate the novelty of the proposed method, in comparison with the following paper that were reported in order to improve fill factor of the stretchable light-emitting devices. In addition, advantages of the proposed method need to be provided. In Ref. 36, hidden pixels based on 3D patterned origami structures have been already demonstrated including full display demos. Hidden pixels are typically required to be turned on at certain threshold stretching conditions as reported in Adv. Mater. 26, 3094 (2014) and IEEE Sensors Journal 20, 14655 (2020). However, if the hidden pixels are always on, the concept is exactly same as Ref. 36 although authors enlarged the light-emitting area by using OLED instead of inorganic LED chips.

2. For quadriaxial stretching approach has been already proposed in 10.1002/adfm.202208792, but it has not been mentioned in the paper. Authors seem to describe the approach is newly proposed in the paper.

3. The proposed method seems to be good only for lighting applications, not for display applications. However, authors mentioned that it is useful for display applications in introduction and discussion by commenting issues of resolution and picture quality. Can it be directly applied for display application? Any idea for electrode design for addressing pixels in an array form? If the method has limitation in its applications, the content of the paper needs to be modified appropriately.

4. Clarification and more discussion must be provided for analysis on OLED properties.

- There are non-uniform brightness distribution depending on amount of Z-axis deformation in figure 5. From the figure and supporting video 1, interconnector area is brighter than the rigid island area when the device is not stretched. Any viewing angle dependent color shift of the cavity-based OLED structure when the device deformed in Z-axis direction.

- In Figure 5(a), current density and brightness increase when the device is 30% stretched. Why? Brightness was measured on island area, bridge area, or the whole devices area including island/bride? Since the hidden area is always on, how did authors deal with it for calculating current density,

luminance, and efficiency before and after the whole area is stretched. Please describe how the current and brightness were measured in detail.

5. Please include conditions of simulation and measurement in the corresponding figures for better understanding of the readers. For example, initial L value must be provided for conventional and proposed platforms in Fig. 1(b). Conditions for biaxial (orthogonal and diagonal) stretching must be provided in Fig 3(b), where authors compared the results with the quadaxial stretching case. Fig 4(e) shows strain distribution for each layer of OLED. Therefore, name (for example, Al, Organic layer, Ag, etc.) and thickness of each layer must be provided in the figure.

6. SR-OLED is not widely used terminology and just made by authors. Instead of SR-OLED, stretchable OLED is recommended to be used.

Reviewer #3 (Remarks to the Author):

This manuscript demonstrated a type of stretchable OLEDs based on a hidden three-dimensional active area. They proposed a 3D architecture adopting a hidden active area that serves a dual role as both an emitting area and an interconnector. This design enables the SR-OLEDs to not only exhibit an initial FF of nearly 100%, but also to maintain their FF even after substantial deformation, demonstrating the efficacy of the proposed approach in realizing the full potential of SR-OLEDs. Overall this work is interesting. However, the reviewer suggests a major revision before it can be published. The comments are as follows:

1. For display applications, the OLED density is very essential for high quality imaging. The device size in the manuscript is several millimeters. How about the scaling of the device? Can this technology achieve the device density of the retina screen?
2. The yield of the device is the basis for the industrial applications. Can this technology achieve high yields for industrial applications?
3. High-speed refresh, 60 Hz or more, for example, is required for display. The characterization of the response speed of the device is missing.
4. How long can the device work stably? Can the device work stably for 10,000 hours to meet the standards of industrial applications?
5. For experimental details, the optimization rationale of 0.6 mm spacing need more evidence and explanations.

Other minor issues:

1. The abbreviation of “Stretchable organic light-emitting diodes” as SR-OLEDs is not proper and understandable.
2. For the description of “two main challenges” of stretchable OLEDs in this manuscript, it can be summarized as one that is “limited initial and post-stretch fill factor”, which results in low resolution for display application as well as limited illumination functions for wearable phototherapeutic applications.
3. Some written mistakes (misusage of it, a/an, what) need to check:

For instance,

- (1) “The HAA being concealed in its initial, unstretched state, it emerges to the surfaces upon stretching, thereby allowing a high FF to be maintained both before and after stretching.”
- (2) “Consequently, $FF^{(p)}$ decreases at a much slower rate with respect to ϵ_{sys} than $FF^{(c)}$, is always higher than $FF^{(c)}$ at a given ϵ_{sys} ...”
- (3) “ana PDMS...”

4. Non-unified format of numbers and parameters (including the typeface and bold or not):

Eg. “ $\nu_E=1.0$, $t_{is}/t_{base}=0.8$ ”.

Eg. Unit of luminance: “cd/m²” in main manuscript while “cd m⁻²” in Supplementary Information.

**Response to Reviewer's Comments and Summary of Changes**

We would like to thank the referee for his or her thoughtful comments and careful review of
our manuscript. The reviewers kindly had several comments and suggestions for improvements.
Responses to each of the comments are summarized as follows:

8 **Reviewer #1's Comment**

This article introduces a novel design for biaxially stretchable OLEDs, aimed at enhancing the
fill factor by employing concealed active areas as stretchable interconnectors. A quadaxial pre-
stretching strategy was implemented on a substrate with meticulously controlled modulus,
resulting in an undistorted island arrangement. Furthermore, OLED configuration simulations
were conducted to mitigate strain on the device. While this study holds promise for high fill
factor stretchable OLEDs, the advancements made, in comparison to existing literature, appear
to be somewhat limited. As a result, concerns may arise regarding its suitability for publication
in Nature Communications.

**Comment #1**

To distinguish this work from previously reported findings, it is essential to include an active
matrix or passive matrix array demonstration. The current setup lacks a substantial
differentiation from previously documented wrinkled or buckled biaxially stretchable LEDs,
bearing a striking resemblance to instances such as "Nat. Commun. 2013, 7, 11573." and "Nano
Lett. 2020, 20, 1526." Hence, the authors are advised to exhibit an array that showcases
individual pixel control capabilities.

**Authors' response to comment #1**

We appreciate the reviewer for highlighting this crucial point. We concur with the reviewer's
suggestion to incorporate a demonstration of the passive matrix array, as it significantly
fortifies the advancements demonstrated in comparison to the two referenced papers and
illustrates how the proposed hidden active area can benefit stretchable displays by not only
providing high fill factor but also enabling resolution compensation. To this end, we have
included the results of the passive matrix array demonstration in **Supplementary Movie 3, 4,**
**5, 6,** and **Fig. 6.** In addition, we have provided comprehensive descriptions of the system setup,
fabrication process, design rules for electrode design, and operational mechanisms in the main
text's experimental section, along with **Supplementary Fig. 14** through **18.**

**The revised part in manuscript is as follows:**

**(1) (Page 3) Introductions in Main text**

[revised manuscript text omitted]

**(5) (Page 17 in Supplementary Information) Supplementary Figure 14**

**Supplementary Figure 14 | System setup for demonstration of a stretchable passive matrix (PM) display:** (a) Schematic
diagram and (b) photograph of the system setup.

(6) (Page 19 in Supplementary Information) Supplementary Figure 15

**Supplementary Figure 15 | Shadow masks for PM stretchable OLED deposition.** (a) Schematic diagrams of the guide
mask and shadow masks defining the patterns of each layers in the proposed stretchable PM OLED device. The guide mask is
used to securely hold the shadow masks for tight control of alignment margin. (b) Photograph of the guide mask and shadow
masks.

(b)

(7) (Page 20 in Supplementary Information) Supplementary Figure 16

Supplementary Figure 16 | The top views of a deposited area through the shadow masks for each of the layers in OLEDs: (a) anode, (b) organic, (c) cathode, (d) bus electrode, and (e) capping and encapsulation layers. (f) Photograph of the fabricated stretchable PM OLED. Note that the substrate in (a)-(e) is the ultrathin parylene before integration onto the 3D patterned elastomer. The photograph in (f) is for the sample integrated onto the patterned elastomer. It is in a stretched state.

**(8) (Page 21 in Supplementary Information) Supplementary Figure 17**

**Supplementary Figure 17** | The conceptual diagrams depicting the ‘D’, ‘I’, ‘S’, ‘P’, ‘L’, ‘A’, and ‘Y’ alphabets through PM
stretchable OLEDs: conventional devices in (a) the initial and (b) the stretched state without hidden pixels; the proposed
devices with hidden pixels in (c) the initial and (d) the stretched state.

**(9) (Page 22 in Supplementary Information) Supplementary Figure 18**

Supplementary Figure 18 | Schematic illustration and timing diagram of PM operation for seven alphabets in (a) the case mimicking the conventional PM operation without hidden pixels (4×4 operation) and (b) the proposed case with hidden pixels. (7×7 operation for resolution compensation)

**Comment #2**

The descriptions of the simulation results in figures 3 and 4 appear overly simplistic. It is
imperative that the authors engage in a comprehensive scientific discussion of the results,
moving beyond mere reporting.

**Authors' response to comment #2**

We appreciate your valuable feedback. We acknowledge the insufficiency of our
comprehensive scientific discussion regarding Figures 3 and 4 and agree that further
quantitative analysis/ discussion is necessary. In response to your feedback, we have revised
our work by providing comprehensive discussion for the mechanical simulation results:

**A. Discussion and revision made for Figure 3**

Beyond simply comparing the quadaxial, orthogonal, and diagonal stretching methods, we
conducted simulations aiming for a quantitative analysis of the ϵ_{diag} effect. In the newly
added **Fig. 3c** and **d**, and **Supplementary Fig. 4**, $\epsilon_{\text{diag}}/\epsilon_{\text{ortho}}$ was gradually increased from
0 to 1 with a step of 0.1, calculating the misalignment (Δy) of islands with respect to the straight
line. The results indicate that, when $0 < \epsilon_{\text{diag}}/\epsilon_{\text{ortho}} < 0.5$, the islands positioned at the four
corners exhibit less y -axial displacement ($\Delta y < 0$) in a parabolic shape compared to the straight
line. At $\epsilon_{\text{diag}}/\epsilon_{\text{ortho}} = 0.5$, Δy approaches 0, reaching a minimum magnitude. As
$\epsilon_{\text{diag}}/\epsilon_{\text{ortho}}$ increases further, however, the islands at the four corners show larger y -axial
displacement ($\Delta y > 0$) compared to the straight line, resulting in increased misalignment in the
shape of an inverted parabola. This discrepancy occurs because the strain applied orthogonally
affects the central islands more than those at the corners, where the strain's impact gradually
diminishes. Hence, compensating for this effect, ϵ_{diag} plays a crucial role, underscoring the
necessity of quadaxial stretching for precise island alignment. In conclusion, further analysis
verified that ϵ_{diag} induces a Δy modulation effect on the islands at the four corners. For the
material combination used (island = 6:1 mixed PDMS, base = EcoflexTM 00-20), the optimal
condition minimizing misalignment corresponds to the case of $\epsilon_{\text{diag}}/\epsilon_{\text{ortho}} = 0.5$.

**The revised part in manuscript is as follows:**

**(1) (Page 7) Results in Main text**

[revised manuscript text omitted]

Figure 3 (Before)

Figure 3 (After)

272 (3) (Page 8 in Supplementary Information) Supplementary Figure 4

Supplementary Figure 4 | The ANSYS simulation results for deformation along the y -axis of the patterned-hybrid elastomer under stretching, at $\epsilon_{\text{diag}}/\epsilon_{\text{ortho}}$ ratios of (a) 0, (b) 0.5, and (c) 1. Plots in (d), (e), and (f) show the magnified view for the results of the y -axis deformation for the area of the 6×6 island arrays in each case.

**B. Discussion and revision made for Figure 4**

Following the advice of Reviewer #1, we have included a further discussion on two types of
stress concentration effects. The first stress concentration effect pertains to the geometry of the
device structure like sharp corners such as point A (A'). The second stress concentration effect
is associated with a significant modulus mismatch between the thin film layers. Upon
examining **Fig. 4e** layer by layer, we have added discussion on irregularly large strains that are
observed at the interface between relatively high modulus materials such as Al (140 GPa), Ag
(70 GPa), Al₂O₃ (134 GPa), and the lower modulus organic layer (2.15 GPa).

In the previous full-area emitting devices, dealing with stress concentration effects at
hinge areas necessitated the use of a very thin (1- μ m-thick) parylene C layer. This was essential
to protect the layers in the full device stack from strain-induced crack formation, but it made
the overall handling process more challenging. However, in the PM OLED demonstration, the
emission area does not extend across the hinge region. Consequently, as long as the strain on
the electrodes is below their COS near the hinge, it should not pose a problem. This reduced
requirement broadens the option for substrate thickness so that we can select a thicker substrate
for easier handling during the integration process. This aspect has also been discussed in the
Section titled 'Fabrication and demonstration of passive matrix stretchable OLEDs' under
Methods.

**The revised part in manuscript is as follows:**

**(4) (Page 10) Results in Main text**

(Strain analysis of mechanically brittle layers in HAA) ... The strain at the bottom and top
Al₂O₃ layers, estimated by varying the thickness of the Parylene C (t_{pary}) from 1 μ m to 9 μ m,
are presented in **Fig. 4b** and **4c**, respectively. Due to the unique structure of the HAA, where
both ends are fixed to two adjacent islands and buckling occurs in between, a significant strain
concentration effect occurs at point A (A'), resulting in the largest strain thereon. **This strain**
**concentration occurs due to the sharp corner that can be found at point A (A'). The level of**
**strain concentration at the corner varies depending on the stiffness of a given structure, which**
**is determined by its geometric parameters.⁵⁴ The simulation results indicate that the strain**
**concentration effect at the point A (A') becomes more severe as t_{pary} increases, because higher**
**t_{pary} yields a structure with higher stiffness. Upon closer examination, the values of strain**
**tensors in the x , z , and shear directions at the point A (A') consistently increase with t_{pary} .**
**(Supplementary Fig. 6) Specifically, the dominant factors contributing to the overall increase**
**in strain with thicker Parylene C layers were identified as the increase in x -axial strain tensor**
**and shear strain tensor. This is because the HAA structure, originally expanded in $\pm x$ -axis**
**direction, is compressed in $\mp x$ -axis direction, causing a restoring force that tends to increase**
**strain tensor elements along x - and shear direction.**

**(5) (Page 11) Results in Main text**

(Strain analysis of mechanically brittle layers in HAA) ... Even with the relatively large strain
applied to A and C, it was confirmed that the proposed structure ensures that the strain applied
to the bottom and top Al₂O₃ in both A and C is below 0.3%. As shown in Fig. 4e, the strain
distribution at point C exhibits a typical profile that is proportional to the distance from the
neutral plane of each layer, without any specific peaks.⁴⁷ Upon closer examination of Fig. 4d
and Fig. 4e, however, another strain concentration is observed at point A at the interface of
Al/organic and organic/Ag/capping layers, attributed to the significant modulus mismatch
between the layers.⁴² Nevertheless, the strains on the anode (Al) and cathode (Ag) layers are
still less than 1% and thus remain below their respective COS values, justifying the use of the
proposed design in terms of reliable HAA operation.

**(6) (Page 17) Methods in Main text**

(Fabrication and demonstration of passive matrix stretchable OLEDs) ...

As previously discussed in Fig. 4, designing Parylene C with a thickness of 1 μm
significantly reduces the strain concentration effect, ensuring mechanical stability. However,
an excessively thin Parylene C thickness might lead to drawbacks in the manufacturing process,
such as substrate tearing or challenges in the transfer process, causing lower yields. Unlike the
previous full-area device, in the case of the PM OLED, the active area does not reside at the
hinge part (Fig. 4, point A (A')). Consequently, there is more flexibility in the design of
Parylene C thickness as long as the strain imposed on the electrode layers remains below the
COS. Therefore, to leverage the advantage of enhancing the process yield, a 3 μm-thick
Parylene C layer was employed. At this thickness, the strain applied to points A (A') and C (C')
remains below 1%, allowing for a sufficiently stable design to protect the electrodes (Fig. 4b,
4c).

**Comment #3**

In the present configuration where the island remains entirely affixed to the substrate, there is
a potential for uniaxial stretching to subject the interconnections perpendicular to the stretching
direction to greater stress due to the substrate's Poisson's ratio. The reviewer raises a concern
regarding whether this could result in device failure, or if the quadaxial pre-stretching approach
is effective in averting such a situation.

**Authors' response to comment #3**

We thank Reviewer #1 for this insightful comment. As this platform was primarily designed
for biaxial stretching, the island array perpendicular to the stretching direction compresses
during uniaxial stretching due to the substrate's Poisson's ratio. This tends to induce a greater
strain along the direction orthogonal to that of stretching. To quantitatively describe the
behavior of the stretchable OLED under uniaxial stretching, we have utilized both ANSYS
simulation (advantageous for describing the behavior of large-dimension elastomers) and
COMSOL simulations (suitable for analyzing thin film structures with large aspect ratios), and
tried to match their results.

The results revealed that when stretched to $\varepsilon_{\text{sub}} = 10\%$, the distance between the
islands reached a geometrically limit (**Supplementary Fig. 13a**) beyond which different parts
of the HAA begin to touch each other. Furthermore, we have analyzed a case where an
additional compression of the islands from their initial state by 0.1mm, mimicking what could
happen during the uniaxial stretching (**Supplementary Fig. 13b**). It was confirmed that, as the
bending radius at point (C) decreased, there is a higher strain observed at point (C), in line with
the reviewer's concern. However, it was found that both top and bottom Al_2O_3 layers still have
strains below 0.3%. These observations confirm that the device can endure uniaxial stretching
of a substrate by 10% on both sides in the uniaxial direction while maintaining stability.

We have added the discussion describing the implication of this uniaxial stretching for
the operation of the proposed stretchable OLEDs.

**The revised part in manuscript is as follows:**

**(1) (Page 14) Results in Main text**

(Performance of the proposed stretchable OLEDs under various mechanical deformations) ...
It was further demonstrated that the device was well operable when it was attached to human
body parts, such as the back of hands and elbows, or when it was stretched using hands in
arbitrary biaxial directions (**Supplementary Fig. 12**). **As the proposed stretchable OLEDs are**
**designed primarily for biaxial stretching, uniaxial stretching could render the interconnections**
**perpendicular to the stretching direction subject to a relatively large stress due to the substrate's**
**Poisson's ratio. Our analysis indicates that the devices are still operable under uniaxial**
**stretching with a substrate strain of up to 10%, although the value higher than that could make**
**devices face geometrical limitation. (Supplementary Fig. 13 for further details)**

**(2) (Page 16 in Supplementary Information) Supplementary Figure 13**

Supplementary Figure 13 | Strain analysis simulation under uniaxial stretching. (a) The ANSYS simulation results for y -axis deformation of the patterned-hybrid elastomer under uniaxial stretching with a 10% substrate strain. ($\epsilon_{\text{sub}} = 10\%$) in x -direction. Due to the substrate's Poisson's ratio, the island array is compressed perpendicular to the stretching direction, causing the distance between the islands to be approximately 0.2 mm closer compared to the initial state. As the distance between the islands decreases to less than 0.2 mm, different parts of the HAA can adhere to each other, limiting further stretching in the uniaxial direction. (b) The COMSOL simulation result illustrating the geometrical change along y -direction before and after applying $\epsilon_{\text{sub}} = 10\%$ in the x -direction. The vertical cross-section is shown for the section indicated as $\overline{PP'}$ in (a). (c) The simulated equivalent strain of the bottom Al_2O_3 and top Al_2O_3 are depicted along the x -axis for both the initial and compressed cases.

Comment #4

Numerous errors have been identified in the references, necessitating thorough proofreading and correction.

Authors' response to comment #4

We thank Reviewer#1 for keen observation. We have thoroughly proofread and corrected any typos or grammatical errors in the references.

**Reviewer #2 (Remarks to the Author):**

This paper reports 3-dimensional structure containing hidden active area by combining
ultrathin OLED sheet and elastic pillar array in order to solve technical issues of resolution
reduction and fill factor lowering under deformation, for stretchable light-emitting devices
based on an island-bridge structure. The fabricated devices showed high fill factors of 97 and
87% before and after 30% stretching deformation and excellent reliability after 1000-time
deformation. Authors well demonstrated stable stretchable devices on various curved and body
surfaces and logical analysis on fabrication process and mechanical stability of the proposed
structures using FEM simulation. After provide some of the issues described below, the paper
is good for publication in Nature Communications.

**Comment #1**

Differentiate the novelty of the proposed method, in comparison with the following paper that
were reported in order to improve fill factor of the stretchable light-emitting devices. In
addition, advantages of the proposed method need to be provided. In Ref. 36, hidden pixels
based on 3D patterned origami structures have been already demonstrated including full
display demos. Hidden pixels are typically required to be turned on at certain threshold
stretching conditions as reported in Adv. Mater. 26, 3094 (2014) and IEEE Sensors Journal 20,
14655 (2020). However, if the hidden pixels are always on, the concept is exactly same as Ref.
36 although authors enlarged the light-emitting area by using OLED instead of inorganic LED
chips.

**Authors' response to comment #1**

We appreciate Reviewer's valuable comment. As pointed out by Reviewer, the introduction
needs further discussion on the novelty and advances from the prior arts. We agree with
Reviewer#2 that the concept of hidden active area and resolution compensation is not
completely new and that we owe to those who pioneered these technologies.

We believe the most significant differentiator of this work (and its advancement
beyond prior art) lies in the fact that, by utilizing the characteristics inherent to organic
technologies, we have achieved both initial and post-stretch fill factors of the active layer at an
unprecedentedly high level. The initial fill factor reaches as high as 97%, and even the post-
stretch fill factor remains close to 90%. Such achievement was possible mainly because we
used OLEDs, whose thin-film, areal-emitting characteristics enabled almost a full coverage of
active layers including hidden areas. It must be noted that transitioning to OLEDs from LEDs,
which most previous works are based on, is not a straightforward task. It requires well-
coordinated strategies tailored to the mechanical characteristics of OLEDs and their process
constraints. This work not only provides architectures and material compositions optimal for
organic devices, but also fabrication processes and strategies to achieve the aforementioned
goals.

Furthermore, passive-matrix display demonstration has been added, and resolution
compensation scheme has also been illustrated in this revision. In fact, comparison made in the
following **Table R1** (shown in the next page) indicates it is challenging to achieve the very
high initial/ post-stretch fill factor and resolution compensation simultaneously, further
corroborating the novelty and advances of the present work. In the revised manuscript,
Introduction has been modified to reflect the discussion made above, and additional references
have been added accordingly.

**Table R1.** Comparison of the FF and the advantages and disadvantages of the high FF
 stretchable electronics platform.

Ref.	Key idea	FF_0 (%) [*]	FF (%) (@ ϵ_{sys}) [*]	Stress-relieving parts	Display demo	Resolution compensation	Key advances or limitations
This work		97	87 (30%)	Hidden "fold-in" active area	Yes	Yes	Both initial and post-stretch FF are high, and resolution compensation is also available.
32 (when all four layers are "on")	N/A	Serpentine interconnectors	Yes	Could be done (at the sacrifice of FF_0)	Low post-stretch FF
17**	9**	Serpentines with added TFTs on them	No (TFT device)	No
23	10 (10%)	Serpentines located beneath the LED array	Yes
34	14 (60%)	Hidden "fold-in" interconnectors	No (PV device)
70	49 (20%)	Hidden "fold-in" interconnectors	No (PV device)
8	8 (100%)	Origami interconnector + Hidden pixel	Yes	Yes
1	3 (40%)	Ag NPs ink printed on Ag electrodes	No		Resolution compensation is available, but FF is low for both initial and post-stretch cases.
2	5 (Uniaxial, 20%)	Ag NPs ink printed on Ni-PDMS composite

 *Estimated value based on the reported device geometry.
 **Even though TFT devices on serpentine are replaced with LEDs, achieving high FF remains difficult due to the
 empty space associated with common serpentine structures.

**The revised part in manuscript is as follows:**

**(Page 2) Introductions in Main text**

... Even for non-display uses, the limited initial and post-stretch fill factor (FF), i.e., the
proportion of the active area to the entire surface area before and after stretching, could be an
issue. For instance, if used for a wearable phototherapeutic patch, the limited FF could result
in some areas of the skin being left unilluminated.

In this work, we propose a novel solution to **achieve rigid-island-based stretchable**
**light sources with the initial and post-stretch FF that are both unprecedentedly high.** To this
end, we exploit the high degree of mechanical flexibility **and areal light-emitting characteristics**
of ultrathin OLEDs in a three-dimensional structure. Specifically, we **replace serpentine**
**interconnectors with an ultra-thin** hidden active area (HAA). **It is hidden in the initial**
**unstretched state by ‘folding inward’** along the negative z axis between adjacent islands of a
3D rigid island array **at a very tight bending radius, thereby allowing a very high FF to be**
**achieved in the initial, unstretched state.** This ultrathin HAA emerges to the surfaces upon
stretching, **alleviating the typical sharp drop in FF upon stretching or, optionally, offering a**
**means to compensate stretching-induced resolution decrease.** Unlike previous approaches
**adopting compact initial integration structures,^{34,35} hidden interconnectors,³⁶⁻³⁸ or hidden active**
**devices,³⁹⁻⁴¹ the HAA in the present work is a part of a whole ultrathin OLED transferred to**
**the islands, and thus, its full area can ultimately function as an active light-emitting region,**
**enabling very high FF even in the stretched state.** We describe a fabrication process to integrate
ultrathin OLEDs onto the 3D array structure, along with a scheme for precise alignment
between these two elements. In particular, we **adopt** a quadaxial stretching method⁴² that
ensures distortion-free alignment, addressing the shortcomings of conventional biaxial
stretching techniques. To implement the HAA concept in a fail-safe manner, we have carefully
designed the device structure and dimensions, using mechanical simulations based on finite
element method (FEM). This ensures minimal strain on the 3D island arrays as well as that on
the encapsulation layers and electrode layers of the ultrathin OLED. The resulting **stretchable**
OLEDs not only exhibit an initial FF of nearly 100% but also maintain their FF up to ca. 90%
of the initial value after experiencing significant biaxial deformation with a system strain of
30%. This is a level of performance unattainable **in conventional 2D rigid-island configurations.**
Furthermore, these high FF stretchable OLEDs demonstrate robustness, with a decrease of only
about 10% in current efficiency even after 1000 biaxial stretching cycles under ca. 30% biaxial
system strain. **Based on this mechanical reliability,** we demonstrate the versatility of these
stretchable OLEDs by showcasing their function on various curved surfaces, including spheres,
cylinders, balloons, and human body surfaces. **Finally, we demonstrate a passive matrix (PM)**
**array utilizing HAA as hidden pixels, capable of compensating for post-stretch resolution**
**reduction inherent to conventional stretchable displays.**

**Comment #2**

For quadriaxial stretching approach has been already proposed in 10.1002/adfm.202208792,
but it has not been mentioned in the paper. Authors seem to describe the approach is newly
proposed in the paper.

**Authors' response to comment #2**

We thank Reviewer #2 for this thoughtful comment. We agree that our initial wording could
give readers impression that the quadaxial stretching system were completely new. We have
included the reference suggested by the reviewer and made the necessary word modifications.

**The revised part in manuscript is as follows:**

**(Page 2) Introductions in Main text**

... We describe a fabrication process to integrate ultrathin OLEDs onto the 3D array structure,
along with a scheme for precise alignment between these two elements. In particular, we **adopt**
a quadaxial stretching method⁴² that ensures distortion-free alignment, addressing the
shortcomings of conventional biaxial stretching techniques.

**Comment #3**

The proposed method seems to be good only for lighting applications, not for display
applications. However, authors mentioned that it is useful for display applications in
introduction and discussion by commenting issues of resolution and picture quality. Can it be
directly applied for display application? Any idea for electrode design for addressing pixels in
an array form? If the method has limitation in its applications, the content of the paper needs
to be modified appropriately.

**Authors' response to comment #3**

We thank Reviewer #2 for this insightful suggestion. As pointed out by the Reviewer #2, the
demonstrations made in the original manuscript were primarily focused on lighting
applications. To reflect Reviewer's suggestion, we have designed and realized a passive matrix
display capable of compensating for resolution decrease based on the proposed platform.
Details on electrode design and addressing schemes, etc. have been added, too. Please refer to
our response to Reviewer #1 - Major Comment #1, as it addresses the same question.

**Comment #4**

Clarification and more discussion must be provided for analysis on OLED properties.

- (1) There are non-uniform brightness distribution depending on amount of z-axis deformation
in figure 5. From the figure and supporting video 1, interconnector area is brighter than the
rigid island area when the device is not stretched. Any viewing angle dependent color shift of
the cavity-based OLED structure when the device deformed in z-axis direction.

- (2) In Figure 5(a), current density and brightness increase when the device is 30% stretched.
Why? Brightness was measured on island area, bridge area, or the whole devices area including
island/bride? Since the hidden area is always on, how did authors deal with it for calculating
current density, luminance, and efficiency before and after the whole area is stretched. Please
describe how the current and brightness were measured in detail.

**Authors' response to comment #4**

We thank Reviewer #2 for keen observation. Our answers to each of his or her questions are
summarized as follows:

(1) Reason why interconnector area appears bright in the non-stretched state and the possibility
of color shift when deformed along the z-direction

The enhancement near the folded HAA is due to the rays emitted from the surface of the HAA
in the initial state undergoing multiple reflections within the folded structure of the HAA,
intensifying the brightness. This phenomenon can be explained through ray analysis, as
depicted in the schematic of **Supplementary Fig. 9a**.

**Supplementary Fig. 9a** further shows that the ray reaching the top from the folded
HAA region shows a prevalence of high-angle emission. This explains the observation that the
CIE x shifts from 0.71 (at island) to 0.58 (at HAA), and the CIE y changes from 0.29 (at island)
to 0.39 (at HAA). This tendency aligns with the angular spectrum's CIE coordinate
characteristics measured with the reference OLED in **Supplementary Fig. 9b and c**; CIE (x, y)
of spectrum at $0^\circ = (0.68, 0.31)$ and CIE (x, y) of spectrum at $80^\circ = (0.58, 0.38)$. In general, the
top emission OLED structure could exhibit a significant viewing angle-dependent color shift
due to the microcavity effect in which the resonant wavelength tends to blue-shifted as the
observation angle increases. It is noteworthy that this angular color shift can be suppressed to
a significant degree with a slight compromise in efficiency through a design where the resonant
wavelength in a normal direction is slightly blue-shifted from that of the peak emission
wavelength, as we demonstrated previously.^{R1} (Please refer to **Fig. (d)** below for detailed
examples)

Figure R1 from reference [R1]. The Fabry-Perot resonance curves (solid lines, $f_{FP}(\theta, \lambda)$) with respect to $I_{EM}(\lambda)$ (dashed line) [left] and the value of $I_{out}(\theta, \lambda) = f_{FP}(\theta, \lambda)I_{EM}(\lambda)$ [right], to which the EL angular radiant intensity is approximately proportional. $I_{EM}(\lambda)$ refers to the free-space PL intensity of an emitter. Results in (b)-(d) are shown for a red OLED over the observation angles (θ) of 0° , 20° , 40° , 60° for the devices that are (b) cavity-resonant, (c) red-detuned, and (d) blue-detuned in a normal direction ($\theta = 0^\circ$).

(2) Regarding the questions on **Fig. 5a**: why do current density and brightness increase when
the device is 30% stretched. Add details on the measurement.

It is considered to result from the fact that the system strain of 30% corresponds to a “flat”
situation where there is no stress associated with folded interconnector region. In other words,
$\varepsilon_{\text{sys}} = 30\%$ represents the state before any mechanical deformation, whereas at 20%, 10%,
and 0%, the HAA gradually folds, causing finite but unavoidable degradation due to
mechanical deformation, resulting in a slight decrease in current density and brightness. The
luminance (L) measurement of the stretchable OLED was conducted at a *single point on the*
*island* where z -axis deformation does not occur, as shown in the inset figure of **Supplementary**
**Fig. 7**, utilizing the imaging spectroradiometer called ‘CS 2000’ (Konica-Minolta), unless
noted otherwise. For calculation of the current density (J), the current that flows through the
whole device was used and divided by the whole active area including that of the interconnector
area. As L and J are both area-normalized values, calculation of L over J provides current
efficiency in cd A^{-1} , even though the device area used for L and J are different.

**The revised part in manuscript is as follows:**

**(1) (Page 11) Results in Main text**

(Performance of the proposed stretchable OLEDs under various mechanical deformations) ...
Additionally, the HAA parts rose to the surface as the device was stretched, turning the
interconnector area, which would otherwise be optically inactive, into an active lighting region,
thereby minimizing the FF loss induced by stretching. (Inset photographs in **Fig. 5b**)

Furthermore, apart from the degradation caused by the mechanical deformation of
stretchable OLED, non-uniform brightness and color distribution, depending on the amount of
z -axis deformation, are observed in the inset images of **Fig. 5b**. Specifically, the interconnector
area appears brighter than the rigid island area when the device is not stretched. This
phenomenon can be explained as light concentration, as depicted with ray diagrams in
**Supplementary Fig. 8a**. In the initial state, rays emitted from the surface of the HAA
undergoes multiple reflections within the structural elements of the HAA, resulting in a light
output through the narrow, confined space and thus making the effective luminance enhanced
locally. If this non-uniformity should be avoided in target applications, matrix addressing may
be introduced so that the light output in the HAA region may be adjusted separately depending
on the strain level.

In addition, it is also observed that, in the non-stretched state, light from the HAA
region exhibit a spectrum that is shifted from that observed in the central region of the rigid
islands. This is because light from the HAA region tends to exhibit relatively high-angle
emission with respect to the emitting surface. In **Supplementary Fig. 8b** and **8c**, the angle-
dependent variations in EL spectrum and CIE color coordinates (x, y) values measured with the
planar, reference OLED are shown. The significant changes in the spectrum and CIE values
are attributed to the viewing angle-dependent color shift characteristic of the 2nd cavity top
emission OLED structure. This observation is consistent with the results of the CIE color
coordinates (x, y) values measured for the proposed stretchable OLEDs, as shown in

**Supplementary Fig. 8;** CIE x changes from 0.71 (at island) to 0.58 (at HAA), while CIE y
changes from 0.29 (at island) to 0.39 (at HAA). Nevertheless, the viewing angle-dependent
color shift can be suppressed to a significant degree through a careful microcavity design only
with a slight sacrifice in the efficiency.⁵⁶

(2) (Page 17) Methods in Main text

(Electro-optical performance evaluation of stretchable OLED under mechanical deformation)
... The electro-optical performance of the stretchable OLED device, encompassing both
current density (J) and luminance (L), was assessed using an imaging spectrophotometer (CS
2000, KONICA MINOLTA, Inc.) in conjunction with a programmable source meter (Keithley
2400) under atmospheric conditions. The current efficiency (η_{CE}) was determined based on the
measured J and L . The CIE color coordinates across the entire surface of the OLED were
evaluated using a 2D color analyzer (CA 2000, KONICA MINOLTA, Inc.). And the angular
electroluminescence (EL) spectra were measured with a customized measurement system
consisting of a fiber optic spectrometer (BW_UVNB, StellarNet) held on a motorized
goniometer (PRM1/MZ8, Thorlabs) for angle-resolved measurements.

(3) (Page 11 in Supplementary Information) Supplementary Figure 7

**Supplementary Figure 7 | The current efficiency (η_{CE}) characteristic versus luminance for the different biaxial system**
**strain (ϵ_{sys}).** (inset: the black dot indicates the measurement spot where L was measured using CS 2000 (Konica-Minolta))

**(4) (Page 12 in Supplementary Information) Supplementary Figure 8**

Supplementary Figure 8 | (a) Schematic diagram illustrating the light ray paths within the HAA in the initial and stretched states. (b) The distribution of angular spectra of the reference OLED having the same structure as used in the device fabrication. (Inset: the photograph of the reference OLED) (c) The angle-dependent change in CIE x , y coordinates of the reference OLED.

**(5) (Page 13 in Supplementary Information) Supplementary Figure 9**

**Supplementary Figure 9** | (a) Setup for measuring the CIE color coordinates of the entire OLED surface area using CA 2000

(Konica-Minolta) (b) Contour plots showing the color coordinates of the stretchable OLED under each system strain.

(a)

(b)

**Comment #5**

Please include conditions of simulation and measurement in the corresponding figures for
better understanding of the readers. For example, initial L value must be provided for
conventional and proposed platforms in Fig. 1(b). Conditions for biaxial (orthogonal and
diagonal) stretching must be provided in Fig 3(b), where authors compared the results with the
quadaxial stretching case. Fig 4(e) shows strain distribution for each layer of OLED. Therefore,
name (for example, Al, Organic layer, Ag, etc.) and thickness of each layer must be provided
in the figure.

**Authors' response to comment #5**

We thank Reviewer #2 for this thoughtful comment. Following the suggestion, we have
explicitly indicated L_{is} , L_{int} , and L'_{int} at the top of **Fig. 1b**. Additionally, in **Fig. 3b**, we included
the strain conditions used for the simulation in the conceptual diagrams of quadaxial and biaxial
stretching. In **Fig. 4e**, we have added the names and thicknesses of each layer and conducted
further discussion related to the strain concentration effect. (Please refer to our response to
Reviewer #1 - Major Comment #2) With this, we have tried to enhance understanding of
readers by providing the details on the conditions of simulation and measurement in the
corresponding figures.

**The revised part in manuscript is as follows:**

**(1) (Page 4) Introductions in Main text, Figure 1b**

Figure 1 (b) (Before)

Figure 1 (b) (After)

(2) (Page 8) Results in Main text, Figure 3b

Figure 3 (b) (Before)

Figure 3 (b) (After)

Figure 4. FEM simulations of Parylene C sandwiched OLED with a HAA structure (a) A schematic diagram of the proposed HAA structure and the candidates (A, B, C, A', B') for strain concentration. (Inset: A SEM image of the HAA is shown, with a scale bar indicating 100 μm) The simulated equivalent strain of (b) bottom Al_2O_3 and (c) top Al_2O_3 are depicted along the x -axis. (Inset: Magnified graphs of the equivalent strain concentration near part A ($x = 0.1 \text{ mm}$)) (d) Enlarged views highlight the three parts (A, B, C) which indicate the equivalent strains in those areas. (e) The simulated equivalent strain along the z -axis is presented at part A ($x = 0.1 \text{ mm}$) and part C ($x = 0.9 \text{ mm}$). **The numerical values displayed within the regions of each OLED layer denote the thickness of individual layers constituting the OLED, shown in units of nanometers (nm).**

Comment #6

SR-OLED is not widely used terminology and just made by authors. Instead of SR-OLED, stretchable OLED is recommended to be used.

Authors' response to comment #6

 We thank Reviewer #2 for the helpful suggestion. As suggested by the reviewer, we have
 replaced the term "SR-OLED" with "stretchable OLED" throughout the document. Thank you.

**Reviewer #3 (Remarks to the Author):**

This manuscript demonstrated a type of stretchable OLEDs based on a hidden three-
dimensional active area. They proposed a 3D architecture adopting a hidden active area that
serves a dual role as both an emitting area and an interconnector. This design enables the SR-
OLEDs to not only exhibit an initial FF of nearly 100%, but also to maintain their FF even after
substantial deformation, demonstrating the efficacy of the proposed approach in realizing the
full potential of SR-OLEDs. Overall this work is interesting. However, the reviewer suggests
a major revision before it can be published. The comments are as follows:

**Comment #1**

For display applications, the OLED density is very essential for high quality imaging. The
device size in the manuscript is several millimeters. How about the scaling of the device? Can
this technology achieve the device density of the retina screen?

**Authors' response to comment #1**

Thank you for your thoughtful comment. According to Reviewer #3's comment, achieving
retina screen-level density by minimizing pixel size is a pivotal milestone in stretchable
displays for high-quality display applications.^{R2} This milestone, however, is challenging even
for the leading companies working on stretchable displays: for example, although there have
been instances of achieving 200 pixels per inch (ppi) stretchable display, the applicable strain
slightly exceeded only 1%.^{R3} Recently, LG Display realized a 12-inch, 100 ppi stretchable
display that can accommodate a 20% system strain,^{R4} marking the state-of-the-art in this
field.^{R5}

If one has to use each of the island as a *single* pixel in the proposed scheme, the present
approach would also lead to a similar level of ppi (i.e. 100-200 ppi) assuming one can make
the side length of the island to be around $127 \mu\text{m}$ ($=25.4\text{mm}/200$) $\sim 254 \mu\text{m}$ ($= 25.4 \text{ mm}/100$),
which we believe is a reasonable dimension that can be achieved with the proposed technique.

However, it should be noted that each island as well as the hidden active area (HAA) region
can host *multiple* pixels, respectively. This is possible because the device is first fabricated on
a planar substrate and then transferred to an array of PDMS islands on an elastomer. In such a
scheme, a higher level of pixel density corresponding to that of 'retina display' should be within
reach.

**Comment #2**

The yield of the device is the basis for the industrial applications. Can this technology achieve
high yields for industrial applications?

**Authors' response to comment #2**

We thank Reviewer #2 for this insightful comment. As the reviewer pointed out, to industrialize
the idea proposed in this paper, it is crucial to ensure sufficiently high yield through a reliable
manufacturing process. In the typical process of stretchable display production, the transfer
process is identified as the main bottleneck affecting yield. As a notable example, in a recently
reported 3-inch, 50 ppi micro-LED stretchable display, there were significant failures during
the roll transfer of multiple micro-LED units, resulting in decreased yield.^{R6, R7}

In the same context, integration of the ultrathin OLED onto a bottom elastomer with
the island array is likely to pose a challenge. Nevertheless, unlike the case of transferring a
plurality of micro-LED units, the whole device in a form of a *single* film needs to be transferred
in the present approach. We believe this will be advantageous in achieving a high yield in the
end. Furthermore, forming and handling AM OLED displays on a thin polyimide has already
been well established in the industry, making the chance of achieving a high yield even higher.
Careful quad-axial alignment would be a challenge, but it should be doable with modern vision
alignment systems.

**Comment #3**

High-speed refresh, 60 Hz or more, for example, is required for display. The characterization
of the response speed of the device is missing.

**Authors' response to comment #3**

We appreciate your valuable comment. Generally, the human eye perceives smooth video
playback from 20-30 Hz and above without experiencing flickering. For this reason, most
displays operate at a high-speed refresh rate of 60 Hz or more. In this context, it is essential to
characterize the high-speed response of any light-emitting device if it is to be used for display
applications involving motion pictures. 60 Hz or above in display applications is essential.
Therefore, we conducted additional verification through demonstrations to ascertain whether
our developed platform can support high-speed refresh operation. Based on the setup of
**Supplementary Fig. 14**, demonstrations were conducted at two different frequencies (10 Hz
and 100 Hz) to compare the degree of flickering in the PM OLED display. We have added
**Supplementary Movies 7A and 7B**, which illustrate the high FF stretchable PM OLED display
operated at 10 Hz and 100 Hz, respectively (operated with the square wave between 0V and
5V). It can be confirmed that there is no flickering for 100-Hz operation while there is for 10-
999 Hz operation, indicating the proposed PM OLED functions properly at video-compatible
refresh rate.

It is also well established that OLED displays can operate at a frequency of 100 Hz or higher
thanks to their solid-state nature, which is consistent with the result shown here.^{R8-R11}

Supplementary Movie 7B (Screenshot). A high fill factor stretchable PM OLED display operating steadily without flickering at 100 Hz.

**The revised part in manuscript based on comments #1 to #3 is as follows:**

**(Page 14) Results in Main text**

**(A stretchable passive matrix OLED display without resolution decrease)** ... Utilizing
HAA as hidden pixels enables the implementation of an advanced display that compensates for
resolution decrease during stretching. **Fig. 6a** and **6b** represent conceptual diagrams of a
conventional platform without hidden pixels and the proposed platform compensating for
resolution decrease through hidden pixels, respectively. Building upon the previously
implemented stretchable OLED platform, we designed a passive matrix array drive system to
individually operate 40 pixels in a 7×7 array structure, excluding the central 9 voids.
Detailed information about the passive matrix (PM) stretchable OLED is provided in
**Supplementary Fig. 14 to 18** and the Experimental section. Examining the implemented PM
stretchable OLED display, in the case of the conventional display with hidden pixels turned off,
even in the initial state, the legibility of alphabets is poor (**Fig. 6c** and **Supplementary Movie**
**3**). When the display is stretched, discerning the shapes of the alphabets becomes challenging
(**Fig. 6d** and **Supplementary Movie 4**). On the other hand, with the proposed platform that
incorporates hidden pixels, the legibility of alphabets is significantly improved both before
(**Fig. 6e** and **Supplementary Movie 5**) and after stretching (**Fig. 6f** and **Supplementary Movie**
**6**). **This clearly illustrates the immense benefits of the proposed platform in stretchable display**
**applications.**

To apply this platform in advanced display applications, it is important to keep the
compatibility with high pixel-per-inch (PPI) density and attain a high production yield. It
should be noted that, although each island hosts only one pixel in the present demonstration,
each island as well as the hidden active area (HAA) region can host *multiple* pixels, respectively.
This is possible because the device is first fabricated on a planar substrate and then transferred
to an array of PDMS islands on an elastomer. In such a scheme, a higher level of pixel density
for moderate-to-high resolution displays could be within reach. At the same time, the proposed
scheme is regarded advantageous for higher production yield while the conventional
stretchable LED displays often encountered low yields due to high failure rates when
transferring multiple LED units.^{57, 58} It is also noteworthy that the present stretchable displays
inherit the high-speed characteristics of OLEDs,⁵⁹⁻⁶² which makes them readily capable of
video-rate operation. (See **Supplementary Movie 7A and 7B** for 10- and 100-Hz operation of
the proposed stretchable OLEDs in comparison with 10-Hz operation.)

**Comment #4**

How long can the device work stably? Can the device work stably for 10,000 hours to meet the
standards of industrial applications?

**Authors' response to comment #4**

We thank Reviewer #3 for this important comment. While the OLEDs shown in this work is
based on research-grade materials and fabricated in a university lab environment, OLEDs made
with the commercial-grade materials (usually not available for university labs) in a factory-
cleanroom environment are known to be operable far more than 100,000 hours.^{R12} There is no
fundamental reason that the proposed approach would not work with such high-quality organic
semiconductors. One extrinsic factor to consider is an encapsulation barrier as organic
semiconductors and their interfaces with electrodes could be sensitive to ambient air. This will
mainly influence the storage lifetime of the device. The encapsulation layer used in this work
consists of parylene and Al₂O₃, which were shown to provide a sufficient level of protection
against ambient air required for practical applications.^{R13}

**Comment #5**

For experimental details, the optimization rationale of 0.6 mm spacing need more evidence and
explanations.

**Authors' response to comment #5**

We thank Reviewer #1 for this keen observation. When designing the dimensions of this
platform, the goal was to achieve initial FF of over 95% and a post-stretch FF of over 85% at
a system strain of 30%. As mentioned in the Method section, the OLED is deposited on a 40
1082 mm by 40 mm substrate, and the size of the stretched OLED should be smaller than 40 mm by
1083 40 mm. Referring to **Supplementary Fig. 16**, the emission layer is on 4 × 4 islands while the
1084 anode and cathode regions are on 6 × 6 islands. Thus, for the convenience of deposition, the
1085 dimension of the 6 × 6 islands were set to 25.4 mm x 25.4 mm, (1 inch²), and the sum of 6L_{is}
and 5L'_{int} should equate to 1 inch. When L_{is} is set to 2.9 mm, L'_{int} is determined to be 1.6
1087 mm, and, at this dimension, the post-stretch FF is calculated to be 87%, meeting the technical
target. Meanwhile, when L_{is} is set at 2.9 mm and L'_{int} at 1.6 mm, L_{int} to achieve a system
strain of approximately 30% is calculated to be approximately 0.56 mm. Therefore, by setting
L_{int} to 0.6 mm, the initial FF aligns with the design objective at 97%.

Furthermore, as evident from Supplementary Equations (S1) and (S5), minimizing
L_{int} maximizes system strain while increasing FF. However, reducing L_{int} to below 0.6 mm
could result in different parts of the HAA adhering to each other, potentially leading to the
mechanical failure of the initial-state HAA structure. Therefore, we optimized it to 0.6 mm,
which is feasible in the fabrication process while ensuring a stable structure and maximizing
the initial FF. We have added relevant discussion to the main text and provided detailed
discussions in **Supplementary Note 1**.

**The revised part in manuscript is as follows:**

**(1) (Page 5) Results in Main text**

**(Overview of the proposed stretchable OLEDs)** ... With the height of the side walls (t_{is}) set
to 0.8 mm, the HAA region in this state is completely concealed in the inter-island region.
Furthermore, we chose L_{int} to be as small as possible to maximize the initial FF close to unity.
This design, with a 0.6 mm spacing, prevents the HAA from touching the bottom surface of
the inter-island region in the initial, non-stretched state, and ensures that different parts of the
HAA do not adhere to each other. As the device is stretched, the island retains its original
dimensions, but the HAA begins to rise toward the top, revealing its appearance gradually.
When the device is fully stretched, the inter-island region is occupied by the flattened HAA
region (1.6 mm wide), which enables non-disconnected light output in the inter-island region,
which would otherwise be occupied with non-lit serpentine interconnectors. The reduction in
the post-stretch FF compared to the initial FF in this case is due to the square void, which has
an area of $A_{void} = (L'_{int})^2 = 1.6 \text{ mm} \times 1.6 \text{ mm}$, present in each cell. (A detailed
explanation regarding the criteria set for each dimension can be referenced in **Supplementary**
**Note 1**)

**(2) (Page 4 in Supplementary Information)**

**Supplementary Note 1**

**D. Design objectives**

One of the key design objectives in this work is to achieve a $FF_0^{(p)}$ higher than 95% and a
$FF^{(p)}$ higher than 85% at $\varepsilon_{sys} = 30\%$. Since the OLED is deposited on a 40 mm by 40 mm
substrate, the size of the OLED consisting of the 6×6 islands were set to 1 inch (25.4 mm
1125 \times 25.4 mm) for convenience. When $L_{is} = 2.9 \text{ mm}$, L'_{int} is determined as 1.6 mm, achieving
a $FF^{(p)} = 87\%$ at $\varepsilon_{sys} = 30\%$, meeting the technical target. Meanwhile, based on these
determined dimensions, L_{int} is calculated to be approximately 0.56 mm to achieve a system
strain of approximately 30%. Therefore, by setting L_{int} to 0.6 mm, the $FF_0^{(p)}$ becomes 97%,
being consistent with the design objective. Furthermore, as evident from **Supplementary**
**Equations (S1) and (S5)**, minimizing L_{int} maximizes system strain while increasing $FF_0^{(p)}$.
However, reducing L_{int} to below 0.6 mm results in different parts of the HAA adhering to
each other, potentially leading to the mechanical failure of the initial state HAA structure.
Therefore, L_{int} of 0.6 mm may be regarded optimal, as it is feasible in the fabrication process,
ensures a stable HAA operation, and maximizes the $FF_0^{(p)}$.

**Minor comments #1**

The abbreviation of “Stretchable organic light-emitting diodes” as SR-OLEDs is not proper
and understandable.

**Authors’ response to minor comments #1**

We thank Reviewer #3 for the helpful suggestion. Please refer to our response to Reviewer #2
- Major Comment #6, as it addresses the same question.

**Minor comments #2**

For the description of “two main challenges” of stretchable OLEDs in this manuscript, it can
be summarized as one that is “limited initial and post-stretch fill factor”, which results in low
resolution for display application as well as limited illumination functions for wearable
phototherapeutic applications.

**Authors’ response to minor comments #2**

We appreciate your valuable comment. In line with the reviewer's suggestion, we have revised
the "two main challenges" to be the "main challenge" for a clearer context in the introduction.

**The revised part in manuscript is as follows:**

**(Page 2) Introductions in Main text**

... This setup allows the active area to be sealed with materials such as alumina (Al_2O_3) and
nitrides, which are one of the best in gas barrier properties yet have relatively low crack-onset
strain (COS). **However, there is a main challenge to overcome.** For display applications, the
presence of stress-relieving interconnectors (e.g., serpentine-shaped interconnectors) limits the
overall resolution or pixel density, and stretching can further reduce resolution, potentially
deteriorating image quality. However, there **is a main** challenge to overcome. For display
applications, the presence of stress-relieving interconnectors (e.g., serpentine-shaped
interconnectors) limits the overall resolution or pixel density, and stretching can further reduce
resolution, potentially deteriorating image quality. Even for non-display uses, the limited initial
and post-stretch fill factor (FF), i.e., the proportion of the active area to the entire surface area
before and after stretching, could be an issue. For instance, if used for a wearable
phototherapeutic patch, the limited FF could result in some areas of the skin being left
unilluminated.

**Minor comments #3**

Some written mistakes (misusage of it, a/an, what) need to check:

For instance,

(1) “The HAA being concealed in its initial, unstretched state, **it** emerges to the surfaces upon
stretching, thereby allowing a high FF to be maintained both before and after stretching.”

(2) “Consequently, $[[FF]]^{(p)}$ **decreases** at a much slower rate with respect to ϵ_{sys} than
$[[FF]]^{(c)}$, is always higher than $[[FF]]^{(c)}$ at a given ϵ_{sys} ...”

(3) “**ana** PDMS...”

4. Non-unified format of numbers and parameters (including the typeface and bold or not):

Eg. “ $\gamma_E=1.0$, $t_{is}/t_{base}=0.8$ ”.

Eg. Unit of luminance: “ cd/m^2 ” in main manuscript while “ $cd\ m^{-2}$ ” in Supplementary
Information.

**Authors’ response to minor comments #3**

We appreciate your valuable comment. We have had the manuscript thoroughly proofread one
more time including those mentioned above.

**References in response letter**

[R1] Kim, E. *et al.* A systematic approach to reducing angular color shift in cavity-based
organic light-emitting diodes. *Org. Electron.* **48**, 348–356 (2017).

[R2] Lee, Y. *et al.* Advancements in electronic materials and devices for stretchable displays.
*Adv. Mater. Technol.* **8**, 1–37 (2023).

[R3] Wang, P. *et al.* A 200 PPI oval shape stretchable AMOLED display. *Dig. Tech. Pap.* **53**,
524–525 (2022).

[R4] Jung, H. *et al.* High-resolution active-matrix micro-LED stretchable displays. *J. Soc. Inf.*
*Disp.* **31**, 201–210 (2023).

[R5] Hong, J., Yoon, J., Kim, Y. & Lee, C. Embracing stretchable “Form Factor-Free” displays.
*Inf. Disp.* **39**, 6–10 (2023).

[R6] Jang, B. *et al.* Highly stretchable color microLED meta-display without image distortion.
*Dig. Tech. Pap.* **54**, 1137–1139 (2023).

[R7] Jang, B. *et al.* Auxetic meta-display: Stretchable display without image distortion. *Adv.*
*Funct. Mater.* **32**, 1–10 (2022).

[R8] Dawson, R. M. A. *et al.* Impact of the transient response of organic light emitting diodes
on the design of active matrix OLED displays. *Tech. Dig. Int. Electron. Devices. Meet.* 875–
878 (1998).

[R9] Buso, D. *et al.* OLED electrical equivalent device for driver topology design. *IEEE Trans.*
*Ind. Appl.* **50**, 1459–1468 (2014).

[R10] Hoffman, D. M., Johnson, P. V., Kim, J. S., Vargas, A. D. & Banks, M. S. 240 Hz OLED
technology properties that can enable improved image quality. *J. Soc. Inf. Disp.* **22**, 346–356
(2014).

[R11] Nabha-Barnea, S., Gotleyb, D., Yonish, A. & Shikler, R. Relating transient
electroluminescence lifetime and bulk transit time in OLED during switch-off. *J. Mater. Chem.*
*C* **9**, 719–726 (2021).

[R12] Patel, B. N. & Prajapati, M. M. OLED: A Modern display technology. *Int. J. Sci. Res.*
*Publ.* **4**, 2250–3153 (2014).

[R13] Sim, J. H. *et al.* OLED catheters for inner-body phototherapy: A case of type 2 diabetes
mellitus improved via duodenal photobiomodulation. *Sci. Adv.* **9**, 1–14 (2023).

REVIEWER COMMENTS

Reviewer #1 (Remarks to the Author):

The authors have significantly enhanced the manuscript, highlighting the novelty of their work by showcasing a stretchable passive-matrix display with resolution compensation. They substantiate their experimental findings with diverse simulations. Nevertheless, the primary concept of the manuscript, namely the near-unity fill factor, may be somewhat insufficient, particularly in relation to Figure 6, to comprehensively address the scope of the entire manuscript. The reviewer maintains that substantial revisions are necessary for the publication of this article in Nature Communications.

1. The title "near-unity fill factor" appears incongruent with the associated data. Despite the implementation of a 3D active area design, the practical fill factor of the common-electrode-type OLED undergoes a notable reduction, shifting from 97% to 87%. Particularly in the presentation of Figure 6, the comprehensive fill factor of the Passive Matrix OLED is not presented, and it is anticipated to be considerably lower than that of OLEDs illustrated in Figure 1. The reviewer contends that the use of the term "near-unity fill factor" is overly exaggerated. It is imperative to include the fill factor variations in PMOLEDs against external strain. In addition, accordingly, a modification to both the introduction and the title of the manuscript needs to be modified.
2. While this article incorporates numerous ANSYS and FEM simulations, it lacks the presentation of detailed information. The method section or supplementary materials should include relevant parameters (e.g., boundary conditions) and assumptions employed in the ANSYS and FEM simulations for clarity and comprehensiveness.
3. In alignment with the comment of Reviewer 3, it is essential to present concrete evidence regarding the actual long-term stability of stretchable OLEDs.
4. In Figure 5a, the J-V-L characteristics of the stretchable OLED are illustrated. However, there is a notable observation that requires clarification: the gradual increase in both current density and luminance when biaxial strain is applied. An explanation for this phenomenon should be provided to enhance the understanding of the depicted results.

Reviewer #2 (Remarks to the Author):

Authors modified the original manuscript according to reviewers' comments. The paper is now good for publication in Nature Communications.

Reviewer #3 (Remarks to the Author):

The authors addressed the comments suitably.

**Response to Reviewer’s Comments and Summary of Changes**

We would like to thank the referee for his or her thoughtful comments and careful review of
our manuscript. The reviewers kindly had several comments and suggestions for improvements.
Responses to each of the comments are summarized as follows:

8 **Reviewer #1’s Comment**

The authors have significantly enhanced the manuscript, highlighting the novelty of their work
by showcasing a stretchable passive-matrix display with resolution compensation. They
substantiate their experimental findings with diverse simulations. Nevertheless, the primary
concept of the manuscript, namely the near-unity fill factor, may be somewhat insufficient,
particularly in relation to Figure 6, to comprehensively address the scope of the entire
manuscript. The reviewer maintains that substantial revisions are necessary for the publication
of this article in *Nature Communications*.

**Comment #1**

The title "near-unity fill factor" appears incongruent with the associated data. Despite the
implementation of a 3D active area design, the practical fill factor of the common-electrode-
type OLED undergoes a notable reduction, shifting from 97% to 87%. Particularly in the
presentation of Figure 6, the comprehensive fill factor of the Passive Matrix OLED is not
presented, and it is anticipated to be considerably lower than that of OLEDs illustrated in Figure
1. The reviewer contends that the use of the term "near-unity fill factor" is overly exaggerated.
It is imperative to include the fill factor variations in PM OLEDs against external strain. In
addition, accordingly, a modification to both the introduction and the title of the manuscript
needs to be modified.

**Authors’ response to comment #1**

We appreciate the reviewer for highlighting this crucial point. We agree with the reviewer that
the term "near-unity fill factor" could be a misleading word choice, considering that (i) post-
stretch FF is 87%, which is still high but not high enough to be claimed as “near-unity,” and
that (ii) FF is significantly reduced in PM OLEDs to ensure a sufficient margin to prevent
crosstalk between adjacent lines.

Following the reviewer’s suggestions, we have revised the manuscript as follows:

**The revised part in manuscript is as follows:**

**(1) (Page 1) Title**

The title has been changed to avoid use of “near-unity” and to reflect its capability of resolution
compensation in display applications as follows:

“Stretchable OLEDs with near-unity fill factor based on a hidden three-dimensional active area”

→ “Stretchable OLEDs based on a hidden three-dimensional active area for high fill factor and
resolution compensation”

**(2) (Page 1) Abstract**

... A portion of the ultrathin OLED is concealed by letting it ‘fold in’ between the adjacent
islands in the initial, non-stretched condition and gradually surfaces to the top upon stretching.
This design enables the proposed stretchable OLEDs to exhibit a relatively high FF not only in
the initial state but also after substantial deformation corresponding to a 30% biaxial system
strain. Moreover, passive-matrix OLED displays that utilize this architecture are shown to be
configurable for compensation of post-stretch resolution loss, demonstrating the efficacy of the
proposed approach in realizing the full potential of stretchable OLEDs.

**(3) (Page 3) Introductions in Main text**

... To implement the HAA concept in a fail-safe manner, we have carefully designed the device
structure and dimensions, using mechanical simulations based on finite element method (FEM).
This ensures minimal strain on the 3D island arrays as well as that on the encapsulation layers
and electrode layers of the ultrathin OLED. The resulting stretchable OLEDs not only exhibit
a high initial FF (97%), but also maintain their FF up to 87% after experiencing significant
biaxial deformation with a system strain of 30%.

**(4) (Page 14) Results in Main text**

(A stretchable passive matrix OLED display without post-stretch resolution reduction) ... To illustrate
the limitation of conventional rigid-island-based stretchable displays, we first operated the
proposed PM stretchable OLED with its hidden pixels turned off, mimicking conventional
stretchable displays. In this case, not only the initial FF is very low at 18%, but also the post-
stretch FF decreases to 11% when stretched with a 30% biaxial strain. Consequently, even in
the initial state, the legibility of alphabets is rather poor due to its relatively large pixel-to-pixel
distance (Fig. 6c and Supplementary Movie 3). When the display is stretched, the situation
gets aggravated (Fig. 6d and Supplementary Movie 4). On the other hand, with the proposed
platform that incorporates hidden pixels, the initial FF increases significantly to 33%, and the
post-stretch FF increases to 26%, compared to the conventional case. As a result, the legibility
of alphabets is significantly improved due to its initial high pixel density (Fig. 6e and
Supplementary Movie 5) which is kept well even after stretching (Fig. 6f and Supplementary

**Movie 6)** thanks to the pixel-density compensation enabled by HAA. This clearly illustrates
the immense benefits of the proposed platform in stretchable display applications. **It is**
**noteworthy that the fill factor of PM OLEDs that appear relatively low may not be regarded as**
**an intrinsic limitation of the proposed methodology. If industrial-grade equipment such as a**
**vision alignment system is used to secure a very tight alignment margin, it is plausible to**
**achieve a fill factor that is quite comparable to that of the common-electrode type stretchable**
**OLEDs, although it will still be limited to ensure line-to-line isolation for prevention of cross-**
**talk.**

**Comment #2**

While this article incorporates numerous ANSYS and FEM simulations, it lacks the
presentation of detailed information. The method section or supplementary materials should
include relevant parameters (e.g., boundary conditions) and assumptions employed in the
ANSYS and FEM simulations for clarity and comprehensiveness.

**Authors' response to comment #2**

We thank Reviewer #1 for this keen observation. Following the suggestion, we have provided
detailed boundary conditions, parameters, and assumptions for both ANSYS and COMSOL
simulations in the Method Section and Supplementary Materials, respectively, to ensure clarity
for the readers' understanding.

**The revised part in manuscript is as follows:**

**(1) (Page 7) Results in Main text**

(Mechanical simulation for integration via quadaxial stretching and design of patterned hybrid
elastomer) ... In this simulation, we applied orthogonal strain (ϵ_{ortho}) and diagonal strain (ϵ_{diag})
to the island arrays at the same distance from the center of the elastomer for all three methods.
(Detailed boundary conditions and assumptions are noted in the **Methods** section and
**Supplementary Fig. 4**)

**(2) (Page 17) Methods in Main text**

(FEM-based mechanical simulation for patterned hybrid elastomer) ... The Young's modulus
and Poisson's ratio of PDMS were set to 1.2 MPa and 0.49, respectively. The simulation results
were presented as equivalent elastic strain values. In ANSYSTM program, the red-colored
regions shown in **Supplementary Fig. 4** were set as surfaces to be grabbed to initiate and
maintain displacement. The full displacement was divided into 13 steps for simulation. No
additional boundary conditions were applied to other surfaces, and it was assumed that there
was no delamination between PDMS and EcoflexTM.

**(3) (Page 8 in Supplementary Information) Supplementary Figure S4**

Supplementary Figure 4 | Detailed boundary conditions for the ANSYS simulation. (a) Tilt view and side view of the quadaxial stretching simulation. The surfaces to be grabbed to initiate and maintain displacement are highlighted in red, while PDMS and Ecoflex™ are depicted in dark gray and light gray, respectively. (b) Tilt view of the orthogonal biaxial stretching. (c) Tile view of the diagonal biaxial stretching.

(4) (Page 10) Results in Main text

(Strain analysis of mechanically brittle layers in HAA) ... In this respect, we analyzed the equivalent strain distribution of the HAA using the finite element method (the structural mechanics module, COMSOL™ Multiphysics), and further details including boundary conditions and assumptions are noted in the **Methods** section and **Supplementary Fig. 7**.

(5) (Page 17) Methods in Main text

(FEM-based mechanical simulation for OLED layers on hidden active area (HAA)) ... To replicate the HAA structure in the simulation which has downward buckling within two PDMS islands, one PDMS island was gradually moved closer to the other PDMS island from their initial distance of 1.6 mm to 0.6 mm in final with ten sub-steps as shown in **Supplementary Fig. 7**. Pre-described displacement boundary conditions of +500 μm on the left PDMS surface and -500 μm on the right PDMS surface were applied, while the remaining surfaces were assumed to be free. It was assumed that there was no delamination between PDMS and the thin-film OLED.

(6) (Page 11 in Supplementary Information) Supplementary Figure S7

Supplementary Figure 7 | Detailed boundary conditions for the COMSOL simulation. The magnified view on the right shows the appearance of the thin-film OLED bonded to the PDMS.

**Comment #3**

In alignment with the comment of Reviewer 3, it is essential to present concrete evidence
regarding the actual long-term stability of stretchable OLEDs.

**Authors' response to comment #3**

As discussed earlier in the previous round of revision, it is difficult to achieve the long-term
stability of OLEDs in the university R&D environment, although it had already been achieved
for commercialized products by the display manufacturer with commercial-grade materials
(usually not available for university labs) in a factory-cleanroom environment. Nevertheless,
we agree with the reviewer that we would better present the actual stability data of the proposed
OLEDs. In particular, we believe it will be meaningful to compare the operational stability data
from the rigid reference OLEDs and those from the stretchable OLEDs with and without
stretch-release cycles.

Our measurement results indicate that the operational lifetimes of rigid reference devices, the
proposed stretchable OLED devices, and the same but those underwent full 1,000 stretch-and-
release cycles (30% - 0%) all closely match one another within the relative difference of 10%
over a period of 24 hours, illustrating that the proposed design can allow for the operational
stability comparable to that of rigid counterparts even under dynamic operations.

**The revised part in manuscript is as follows:**

**(1) (Page 14) Results in Main text**

(Performance of the proposed stretchable OLEDs under various mechanical deformations) ... After
confirming the mechanical stability at each strain level, the proposed stretchable OLED was
tested for the mechanical reliability under repeated cycles of deformation using a home-made
computer-controlled biaxial cyclic test equipment, as shown in the inset in **Fig. 5d** and
Supplementary Fig. 10. With this set up, we measured J-L-V characteristics under biaxial strain
of 30% after 1, 10, 100, 1000 'full stretch and release' cycles. (**Fig. 5c**) Although *J* and *L* at a
given voltage decreased slightly with the number of cycles, η_{CE} remained almost intact, as
shown in **Fig. 5d**. (See **Supplementary Fig. 14** for the lifetime test, and **Supplementary**
**Movie 1** for actual operation)

**(2) (Page 18) Methods in Main text**

(Electro-optical performance evaluation of stretchable OLED under mechanical deformation)
... The stretch-release cyclic test was carried out with a biaxial system strain of 28.6%,
measuring the device characteristics the CS 2000 after 1, 10, 100, and 1000 cycles. The motor
operation speed was set to 200 mm min⁻¹ with a delay time of 0.1 seconds. **The lifetime test**
**was conducted for the reference OLED as well as for the stretchable OLED before and after**
**the cyclic test. A calibrated photodiode (FDS 100-cal, Thorlabs) was used to measure the**
**relative change in the brightness level as a function of operation time. The reference OLED**
**was fabricated on a 4 cm × 4 cm glass substrate, and all three types of devices were operated**
**under constant current driving conditions corresponding to the initial brightness of 100 cd m⁻².**

(3) (Page 18 in Supplementary Information) Supplementary Figure S14

**Supplementary Figure 14** | (c) Lifetime test of stretchable OLEDs before and after 1000 stretch-release cycles. The result for
a rigid reference device is also shown for comparison. (Under 100 nit driving condition, $LT_{50} \approx 6$ hours in all cases. The inset
shows photographs of the reference OLED operated at 100 cd m^{-2})

(c)

**Comment #4**

In Figure 5a, the J - V - L characteristics of the stretchable OLED are illustrated. However, there
is a notable observation that requires clarification: the gradual increase in both current density
and luminance when biaxial strain is applied. An explanation for this phenomenon should be
provided to enhance the understanding of the depicted results.

**Authors' response to comment #4**

We appreciate the insightful comment from Reviewer #1. As stated in our response to Reviewer
#2's comment #4 in the previous revision, this phenomenon could be attributed to the fact that
the *local* maximum strain applied to layers subject to bending-induced fracture is the highest
when the *system* strain is the lowest. Nevertheless, difference appears rather significant,
considering that the proposed devices withstood quite well the cyclic test that includes a full
swing of the system strain between 0% and 30%.

To better understand its origin, therefore, we have carried out an additional experiment (with
another sample from the same batch) in which the maximum voltage applied is limited to 5.25V,
instead of 6V used in **Fig. 5a**. It was observed that, swept under less harsh conditions up to
5.25V, the degree of degradation got decreased, resulting in less variation in the J - V - L
characteristics with system strain of 0, 10, 20, and 30%, as depicted in **Fig. R1**. This
observation suggests that the variations in J - V - L characteristics are not only mechanical in
nature but also stem from electrical degradation, which could be significant especially when
the maximum voltage applied exceeds a certain threshold.^{R1}

**Figure R1** | The current density and luminance characteristics versus driving voltage up to 5.25V for
the different biaxial system strain (ϵ_{sys}) values, measured in the order of 30%, 20%, 10%, and 0%.

In fact, when measuring the samples for **Fig. 5a**, the OLED was first measured (right after
fabrication) at the system strain of 30%, and then it was measured sequentially at the system
strain of 20%, 10%, and 0%. Consequently, after the initial voltage sweep up to 6V, the device
exhibited a noticeable performance drop, resulting in a set of the J - V - L characteristics that
appear degraded as if it were due to the change in the system strain.

Furthermore, it should also be noted that the characteristic variations can result from a
combination of mechanical, optical, and electrical phenomena, making it difficult to pinpoint
specific causes. For example, variations in the electrical characteristics of contacts could occur
due to the Ag-paste used to connect to the external power source during measurement. While
slight variations in characteristic changes were observed depending on sweep conditions or
samples, the overall conclusion is that they operate well without significant characteristic
changes in all cases.

**The revised part in manuscript is as follows:**

**(1) (Page 11) Results in Main text**

(Performance of the proposed stretchable OLEDs under various mechanical deformations) **Fig.**
**5** illustrates the electrical-optical characteristics of the realized stretchable OLED under various
mechanical deformations. As shown in **Fig. 5a**, it can be observed that the current density (J)
and luminance (L) versus voltage (V) characteristics do not change significantly as the devices
are biaxially stretched, **at ϵ_{sys} varied from 30% to 0% at 10% intervals in each orthogonal**
**direction. ($\epsilon_{\text{sys}} = \epsilon_x = \epsilon_y = 30\%$) (Additional experimental data and a detailed explanation**
**regarding the difference in J - V - L characteristics in the high voltage region are provided in**
**Supplementary Fig. 9) Furthermore, the normalized current efficiency (η_{CE}) values at 100 cd**
**m^{-2} were shown to maintain their initial value (4.5 cd A^{-1}) up to 97% when the device was fully**
**stretched. (Fig. 5b, Supplementary Fig. 7) The inset images also confirm that the OLED**
**driven at constant current of 1 mA under ϵ_{sys} of 0%, 10%, 20%, and 30% operated without**
**any visual degradation. (Scale bar corresponds to 1 cm in length.)**

**(2) (Page 13 in Supplementary Information) Supplementary Figure S9**

**Supplementary Figure 9 | The current density and luminance characteristics versus driving voltage up to 5.25V for the**
**different biaxial system strain (ϵ_{sys}) values, measured in order of 30%, 20%, 10%, and 0%.**

The difference in J - V - L characteristics in the high voltage region in **Fig. 5** is not only
mechanical in nature but also stems from electrical degradation, which could be significant,
especially when the maximum applied voltage exceeds a certain threshold. In experiments
conducted with another sample from the same batch, sweeping under conditions less harsh than
the maximum bias of 6V resulted in a lower degree of variation in the J - V - L characteristics
among those with system strains of 30, 20, 10, and 0%, as depicted in above data.

**Reviewer #2's Comment**

Authors modified the original manuscript according to reviewers' comments. The paper is now
good for publication in *Nature Communications*.

**Reviewer #3's Comment**

The authors addressed the comments suitably.

→ We thank Reviewers #2 and #3 for their thorough review and positive comments on our
manuscript.

**References in response letter**

[R1] Peng, H. *et al.* Coulomb effect induced intrinsic degradation in OLED. *Org. Electron.* **65**,
370–374 (2019).

REVIEWERS' COMMENTS

Reviewer #1 (Remarks to the Author):

The authors have properly revised their manuscripts based on the points raised by the Reviewers, and thus the revised manuscript is now in suitable form for publication.